# Comparison of Statistical Approaches for Modelling Land-Use Change

**Bo Sun and Derek T. Robinson \***

Department of Geography and Environmental Management, University of Waterloo,
Waterloo, ON N2L 3G1, Canada; b22sun@uwaterloo.ca
**\*** Correspondence: dtrobins@uwaterloo.ca

**Abstract:** Land-use change can have local-to-global environment impacts such as loss of biodiversity and climate change as well as social-economic impacts such as social inequality. Models that are built to analyze land-use change can help us understand the causes and effects of change, which can provide support and evidence to land-use planning and land-use policies to eliminate or alleviate potential negative outcomes. A variety of modelling approaches have been developed and implemented to represent land-use change, in which statistical methods are often used in the classification of land use as well as to test hypotheses about the significance of potential drivers of land-use change. The utility of statistical models is found in the ease of their implementation and application as well as their ability to provide a general representation of land-use change given a limited amount of time, resources, and data. Despite the use of many different statistical methods for modelling land-use change, comparison among more than two statistical methods is rare and an evaluation of the performance of a combination of different statistical methods with the same dataset is lacking. The presented research fills this gap in land-use modelling literature using four statistical methods—Markov chain, logistic regression, generalized additive models and survival analysis—to quantify their ability to represent land-use change. The four methods were compared across three dimensions: accuracy (overall and by land-use type), sample size, and spatial independence via conventional and spatial cross-validation. Our results show that the generalized additive model outperformed the other three models in terms of overall accuracy and was the best for modelling most land-use changes with both conventional and spatial cross-validation regardless of sample size. Logistic regression and survival analysis were more accurate for specific land-use types, and Markov chain was able to represent those changes that could not be modeled by other approaches due to sample size restrictions. Spatial cross-validation accuracies were slightly lower than the conventional cross-validation accuracies. Our results demonstrate that not only is the choice of model by land-use type more important than sample size, but also that a hybrid land-use model comprising the best statistical modelling approaches for each land-use change can outperform individual statistical approaches. While Markov chain was not competitive, it was useful in providing representation using other methods or in other cases where there is no predictor data.

**Keywords:** land-use change model; Markov chain; logistic regression; generalized additive model; survival analysis; spatial cross validation

---

## 1. Introduction

A variety of approaches are used in land-change science to represent land-use change (e.g., statistical [1]; cellular automata [2]; agent based [3]). Among the methods used to model land-use change, empirical statistical models are often used to detect the effects of drivers [4] of change as well as to simulate potential quantity and pattern outcomes of land-use change [5]. To date, many statistical

models have been used to model land-use change (e.g., [6,7]), of which logistic regression and linear regression are the most frequent [8]. In some cases, statistical models are combined with other methods, for instance, Markov chain approaches are often coupled with logistic regression (e.g., [9]), cellular automata (e.g., [10,11]) and genetic algorithm (e.g., [12]) to model land-use change.

The benefit of using statistical models arises from their ease of implementation and application as well as their ability to provide a general representation of land-use change given scarce time, resources, and data. Furthermore, statistical models have a longer history and frequency of use, and they are more likely to be used in other research projects that do not have a direct collaboration with their creators compared to process-based models [13]. The trade-off, however, is that statistical modelling approaches poorly represent the explicit processes associated with human decision-making (e.g., farmers' planting decision on agricultural lands), which is better represented by process-based models such as agent-based models [14].

Despite the utility and widespread use of statistical methods for modelling land-use change, there is a lack of review or assessment of the performance of more than two different statistical methods (or different combinations) with the same dataset at the same location. Interesting exceptions to this lacuna include: a comparison of two variants of logistic regression and survival analysis applied to an urban sprawl model (i.e., a controlled setting), which showed superior performance of survival analysis [15]; and a comparison of input, output, and validation maps from nine different land-change models applied to 12 different locations [7]. Results of the nine models were compared against observed change and a null model of persistence from the initial time-step of land-use data. However, the statistical methods behind the presented models, overall model accuracy, or accuracy by land-use type were not discussed. Furthermore, the models were applied to different data and when they were applied to the same location the format of the data were different, which obfuscated their comparison.

Other scientific fields of study have also noted the lack of comparative studies of more than two statistical modelling approaches applied to the same data (e.g., ecology [16]). While some comparisons exist (e.g., species presence/absence [17]; landslide susceptibility [18]), a gap remains in the land-use literature. We seek to contribute to overcoming this gap in the comparative statistical literature on modelling land-use change by comparing the performance of four conceptual approaches (stochastic process, parametric model, non-parametric model, and time series model) to modelling land-use change. These approaches span a range of frequency of application in land-use modelling literature and are commonly operationalized as Markov chain (MC), logistic regression (LR), generalized additive model (GAM), and survival analysis (SA). The four modelling approaches are applied to a common study area using the same data. Their performance is evaluated in terms of their ability to predict overall land-use change as well as change by specific land-use types. Through this analysis we answer the questions: what is the overall accuracy of different statistical methods in representing land-use change, and how well do these methods compare across individual land uses?

## 2. Materials and Methods

### 2.1. Study Area

Our study area is situated in the Region of Waterloo, which comprises 1369 km$^2$ in Southern Ontario, Canada (Figure 1). The region is composed of three cities (Kitchener, Waterloo, and Cambridge) and four townships (Wellesley, Woolwich, Wilmot, and North Dumfries), within which exists a mixture of residential, commercial, agricultural, and other land-use types. The Region has been experiencing above average population growth and subsequent land-use change due in part to employment opportunities in the high-tech research and development sector, low-cost housing relative to the City of Toronto, its location along highway 401 (the busiest highway in North America [19]), and close proximity to the City of Toronto (the 5th largest North American financial center [20]).

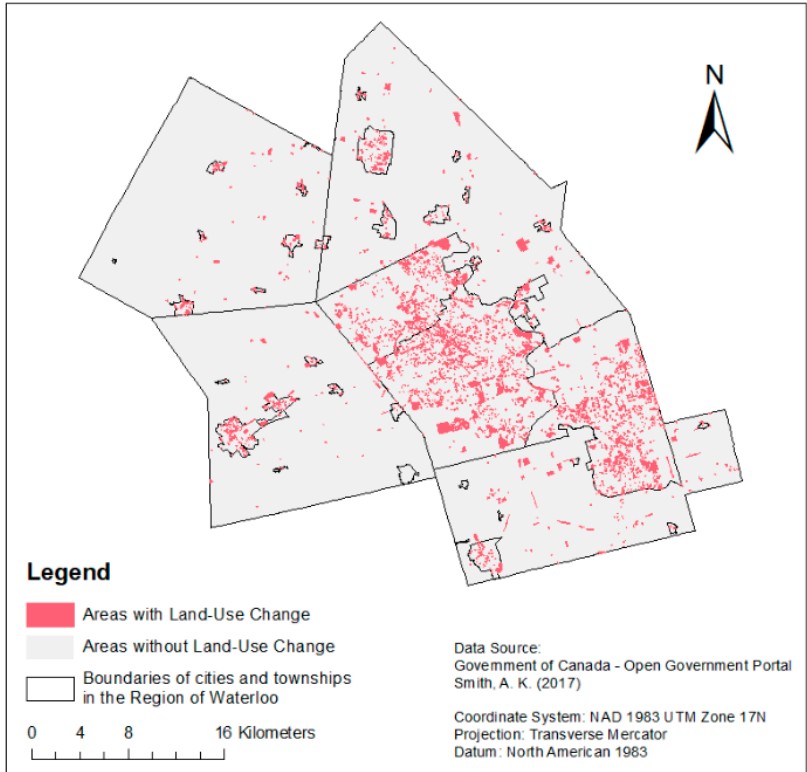

**Figure 1.** Land-use change from 2010–2015 in the Region of Waterloo, Ontario, Canada.

The population of the Region of Waterloo increased from 507,096 to 535,154 from 2011 to 2016, which was a 5.5 percent increase in population that exceeded the 4.6 percent provincial and the 5 percent national population growth rates recorded in 2016 [21]. The Kitchener-Cambridge-Waterloo Census Metropolitan Area had the 4th largest population in Ontario and the 10th largest population in Canada according to the 2016 Census data. Moreover, the population growth rate in the Kitchener-Cambridge-Waterloo census metropolitan area ranked the 4th highest in Ontario in 2016. The fast growth rate for the region has resulted in urban sprawl, which influences the types of land-use transitions in the region.

Between 2010 and 2015, 10,606 of 160,053 (6.63 percent) individual property ownership parcels experienced land-use changes [22] (Figure 1), in which 51 percent of the parcels converted to medium-density residential land use and 16 percent of them converted to commercial land use. Other noticeable land-use changes included changes to transportation (9 percent), high-density residential (8 percent) and a number of properties were found in transition and under development (7 percent). The remaining 9 percent of change were attributed to the remaining six land-use types. Within these overall land-use changes, the dominant transitions were under development to residential (34.24 percent) and commercial to residential (20.77 percent).

*2.2. Data*

Ownership property parcel was used as the observational unit of land-use change in this study, and the data were acquired from Teranet for the Region of Waterloo for the year 2010. Land-cover-raster data were acquired from the Modelling and Spatial Analysis Lab at the University of Waterloo (Waterloo, ON, Canada) [22], which comprised eight land-cover types at a resolution of 80 cm. These land-cover data were used to classify property parcel land-use data for the Region of Waterloo, for the years 2006, 2010, and 2015 [22]. The land-use classes in these data consist of the following ten land-use types and water: low-density residential, medium-density residential, high-density residential,

commercial, industrial, institution, transportation, protected area and recreation, agriculture, water, and under development.

A variety of drivers of land-use change can be found in the literature, which varies among models focused on specific types of land-use change versus comprehensive land-use models (i.e., those representing all types of land-use change in a study area). Often these variables are grouped by similarity and then individually evaluated. For example, conceptual groupings include: population, affluence, and technology [23]; key policies, technology, markets, demography, endowments, infrastructure, and institutions [24]; and demography, economy and infrastructure, and geomorphology [25]. Drivers of land-use change used in this paper were selected among commonly recognized predictors among these groupings, such as mean slope and elevation (e.g., [25,26]) and population density (e.g., [27,28]).

In this paper, drivers of land-use change were conceptualized into geometric variables (e.g., parcel area and area of Dissemination Area), site variables (e.g., slope and elevation), demographic variables (e.g., population density), and spatial variables (e.g., distance from a parcel to the nearest highway ramp). Geometric variables were calculated using ArcGIS. Demographic variables were acquired from the 2011 Census data (Dissemination Areas boundary polygons and demographic data [29]). The Canadian Census is taken every five years and 2011 provided the closest year to 2010. In addition to these drivers, variables were derived from data acquired from the Ontario Ministry of Natural Resources and Forestry (Rivers, Water Bodies, Wooded Areas, and a Digital Elevation Model (10 m resolution)), and the Ontario Ministry of Transportation (Toronto, ON, Canada) (Road Network). A final dataset created by the Modelling and Spatial Analysis Lab represented the access point of the road network to all major 400 series highways in the province of Ontario. In addition to variable creation, variable transformation was performed to unify the units of distance variables from meters to kilometers, units of population density variables from person/$m^2$ to person/$km^2$, units of parcel geometry variables from $m^2$ to $km^2$, and units of elevation from $m^2$ to $km^2$. The measurement transformations ensure the gradients of values of all predictors are on a similar scale. The detailed pre-processing steps of the land-use data are documented in Appendix A, and a full list of variables used in model building can be found in Appendix B.

In addition to the aforementioned data, zoning is typically considered a critical factor that regulates, restricts, and fosters the types and locations of land-use change in a municipality. Thus, a change of land use can be prohibited to occur in an area zoned for a particular land-use type [30]. Despite common zoning policies (e.g., by land-use type or minimum lot size), zoning may vary across municipalities and may change over time within a single municipality, which creates a spatially and temporally heterogeneous set of zoning rules. Furthermore, variations in designated land-use zones may occur [31] and occur with different frequencies by municipality. Due to these inconsistencies in the zoning process among municipalities and the lack of any central repository consolidating the Master Plans of the 444 municipalities of Ontario, zoning was excluded as a predictor variable in the presented statistical modelling approaches. Although the presented research uses the case study of the Region of Waterloo, inclusion of the zoning information as a driver in a statistical model to predict future land-use changes would make the model non-transferable, which means that the model is restricted to predict local land-use changes and contribute to local land-use planning area. Our goal is to construct statistical models that can be applied widely across the province of Ontario and more broadly Canada to represent land-use change patterns.

*2.3. Statistical Modelling Approaches*

2.3.1. Markov Chain

Markov Chain (MC) is a statistical method that incorporates stochasticity in the process of changes between states. A discrete-time MC requires a countable set of states and events that are mutually exclusive and collectively exhaustive [32]. Moreover, uniform length is required between any two time

points. In this research, a discrete-time MC method was used to model land-use changes that occurred between 2010 and 2015 at a parcel level, which is considered a first-order discrete-time MC since the status at a given time only depends on the status of the nearest past state. Using recent land-use data (2015) and the past land-use data (2010), an eleven by eleven transition probability matrix that contains only one-step transition probabilities was calculated. The probability matrix was obtained by observing the frequencies of land-use changes between the two years. To test the performance of MC, a random value between 0 and 1 was assigned to each parcel in the test set to represent the land-use transition probability from 2010 to 2015. For each starting state of land use, a probability space of 0 to 1 was divided into eleven non-overlapping segments based on the eleven one-step transition probabilities in their corresponding rows in the transition probability matrix. The segment within which the randomly drawn parcel value lies determines the future land use of the parcel in 2015. Then, the predicted land uses of all parcels in the test dataset were compared to the real land uses of those parcels in 2015 to determine the performance of MC for making prediction of future land uses.

### 2.3.2. Logistic Regression

Logistic regression (LR) is a type of statistical method used to model categorical variables and is a member of generalized linear models. Its response variable follows a binomial distribution and connects to the linear combination of all covariates though a logit-link function. LR can be used to simulate and predict categorical land-use change outcomes [33]. Multiple LR, which contains more than one independent variable, has often been used to model land-cover change (e.g., [34]). Multiple LR was used in the presented research among all other models in the family of LR (e.g., ordinal LR and multinomial LR). When predicting parcels with unknown land-use types, either 0 or 1 was assigned to each parcel based on the estimated probability and a land-use-change threshold. If the land-use change status at a location was determined to be 1, it means the model predicted the parcel converts from one land-use type to a target land-use type; otherwise, it means no change occurred.

### 2.3.3. Generalized Additive Model

Generalized additive models (GAMs) extend generalized linear models (GLMs) by using a series of smoothing splines to express the non-linear relationship between the expected mean of responses and a set of predictors [35]. GAM has been implemented in land-use science in addition to GLM (e.g., [36]). The advantage of GAM over GLM is that it has the ability to represent non-linear relationships, which ensures that more realistic situations can be represented. However, the effect of a non-linear predictor modelled by a smoothing spline cannot be interpreted using one consistent coefficient. Even though sometimes a smoothing spline can be replaced by a simple function (e.g., square and cubic function) across the entire range of variable values when we increase the degree of smoothness of the smoothing spline, the transformed predictor can be very difficult to interpret compared to linear predictors. The presented research used the additive logistic regression model from the family of GAMs. Additive logistic regression was chosen because its response variables (i.e., statuses of land-use changes) are binary. Similar to LR, a value of 0 or 1 was assigned to each parcel to determine the status of predicted land-use change.

### 2.3.4. Survival Analysis

SA analysis (SA) is used mostly in health and clinical studies to predict the mortality rate or recovery rate of patients (e.g., recovery rate from injury or diseases). SA can handle both time-varying and time invariant variables and can take into account incomplete data. SA models use both the duration of each observation in the experiment and an indicator variable showing the occurrence of the event of interest as response variables. All other potential factors that influence the occurrence of the event are used as covariates to calculate the success or failure rate of the event at the time.

Among many SA techniques, a Cox proportional hazards model [37] was used because it assumes constant effects of covariates on hazards over time, which is consistent with the assumptions of

predictor effects in other models (i.e., LR and GAM). In the context of modelling land-use change, the event of interest is the change from one land use to another between two time points and the failure time of a parcel would be the time that a land-use change occurs. Our land-use data include only two points in time 2010 and 2015 giving two observations in the five-year time period. However, multiple measurements of land use for each parcel in the study period are required to determine a survival analysis failure time. Therefore, a time variable was created by randomly generating integers in the range of 1 to 6 to represent the failure time of each parcel within the 2010 (time = 1) and 2015 (time = 6) period. Other values of time represent years between 2010 and 2015 in an ascending order. Discrete time steps were used instead of continuous time to maintain consistency with land-use data and typical annual time steps used in modelling land-use change.

### 2.4. Analysis

As part of our comparative analysis of different statistical methods for representing land-use change, we also evaluated the effects of sample size on model performance. For LR, GAM, and SA, we first created a 'full dataset' for each land-use type comprising all property parcels with the same land-use type in 2015 (Figure 2a). Binary response variables, Y, were created for each parcel in the full dataset to indicate if the state of a parcel underwent a change in land use from 2010 to 2015 or remained unchanged. Eleven full datasets were created corresponding to the eleven defined land uses in 2015.

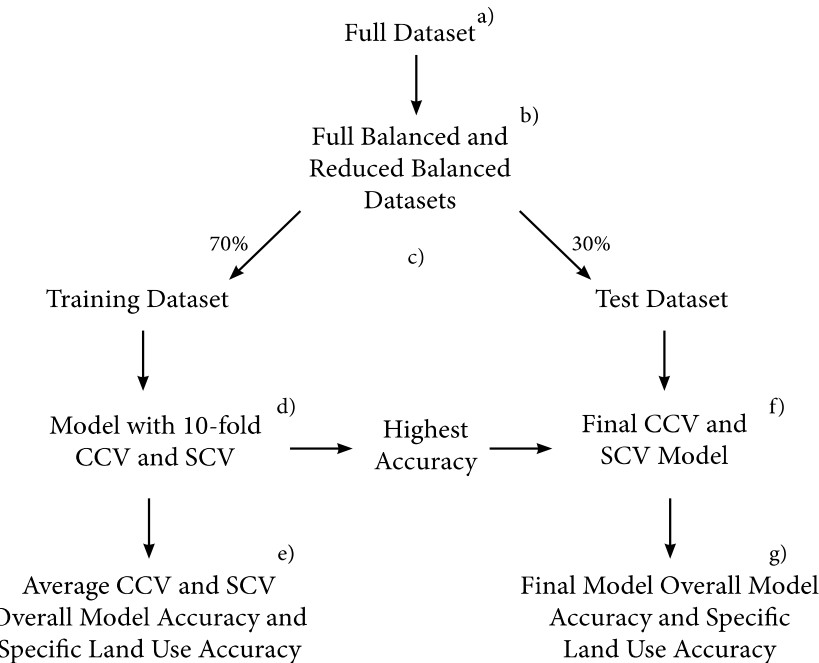

**Figure 2.** Methodology for logistic regression (LR), generalized additive model (GAM), and survival analysis (SA) statistical models including the development of full-balanced and reduced-balanced training and testing data, model selection, and land-use classification accuracy assessment. CCV: conventional cross-validation; SCV: spatial cross-validation.

Each of the eleven full datasets was used to create two new balanced datasets (22 in total), a full and balanced (full-balanced) dataset and a reduced and balanced (reduced-balanced) dataset (Figure 2b). A balanced dataset contains the same number of observations for all existing levels of a categorical independent variable [38], which in this study refers to the dataset containing an equal number of changed and unchanged parcels. One benefit of using a balanced dataset is that it eases the interpretation and use of probabilistic thresholds when predicting change. In this study, when the outcomes of LR, GAM, and SA probabilities of parcel change are greater than or equal to 0.5, the parcel

was classified as changed and otherwise was classified as unchanged. Because LR and GAM produce probabilities and SA outputs can be converted to probabilities, by converting its hazard function to a survival function [6], the three statistical approaches can be easily compared.

A full-balanced dataset corresponding to a specific land use (e.g., industrial) was created by acquiring all parcels that changed to that land use (e.g., industrial) and a matching number of parcels that persisted as that land use over the 2010–2015 time period (e.g., unchanged industrial parcels). The number of changed parcels in a reduced-balanced dataset is the same as the number of unchanged parcels and is set to a maximum of 500 to reduce the potential of over-fitting, yet maintain enough data to construct a model with many predictors and to assess the effects of sample size on model performance. When the number of changed parcels is less than 500 for one type of land-use change, the full- and reduced-balanced datasets for this land-use change are the same. The size of full-balanced datasets ranges from 10 to 10,816 points (Table 1). In our data, no parcel had changed to agricultural land-use from 2010 to 2015. Moreover, only five and thirty-six parcels had changed from some land uses to institution and water, respectively. Thus, there are insufficient data for LR, GAM, and SA to model agricultural, institutional, and water land-use changes with many predictors.

**Table 1.** Sample sizes of full-balanced and reduced-balanced datasets by land-use type.

| Land-Use Change | Full-Balanced | Reduced-Balanced |
|---|---|---|
| Low-Density Residential | 870 | 870 |
| Medium-Density Residential | 10,816 | 1000 |
| High-Density Residential | 1698 | 1000 |
| Commercial | 3320 | 1000 |
| Industrial | 530 | 530 |
| Institution | 10 | 10 |
| Transportation | 1892 | 1000 |
| Protected Area and Recreation | 520 | 520 |
| Agriculture | 500 * | 500 * |
| Water | 72 | 72 |
| Under Development | 1484 | 1000 |

* The samples of agricultural parcels are randomly selected for Markov chain.

The full- and reduced-balanced datasets for each land-use type were partitioned into training (70%) and test (30%) datasets using the traditional hold-out method (e.g., [39]) (Figure 2c). Then, a 10-fold conventional cross-validation (CCV) was applied to the training data to produce an averaged cross-validation accuracy for each land-use type and an overall cross-validation accuracy for each statistical method and the associated full- and reduced-balanced datasets (Figure 2d). A second cross-validation procedure was then conducted using 10-fold spatial cross-validation (SCV, [40]) to determine if spatial autocorrelation was present and had an effect on statistical model performance. Spatial autocorrelation is a common problem in land-use modelling, which violates most statistical model assumptions about independent observations. With CCV, parcels are randomly selected for training and test data, while SCV partitions the parcel data using k-means clustering [41] based on their spatial coordinates. Average CCV and SCV accuracy values for each land-use type and each balanced dataset within the training data were reported and used to calculate an overall model accuracy for the four statistical models evaluated (Figure 2d). Predictor coefficients were estimated using each statistical method and only significant predictors (p-values less or equal to 0.1) were retained from the CCV and SCV that produced the highest accuracy (Figure 2f) for final model development. Final models were fit against the training data and final accuracy values were determined against the test data (Figure 2g).

The datasets used in the MC approach involved combining all full-balanced datasets into one MC full-balanced dataset and all reduced-balanced datasets into one MC reduced-balanced dataset. An additional 500 agricultural parcels were added to each of the MC full-balanced and MC reduced-balanced datasets, which were excluded from the LR, GAM, and SA models due to no

occurrences of change. From MC full-balanced dataset and reduced-balanced dataset, 70 percent were randomly selected for training models with 10-fold CCV and 10-fold SCV and the rest were used for testing the performance of MC. Using these datasets, two transition probability matrices were created as part of the MC modelling process, one for the MC full-balanced dataset and one for the MC reduced-balanced dataset. The probability matrices contain the probability of a land-use transition from each land-use type in 2010 to another land-use type in 2015.

The performance of the statistical models was compared using the overall cross-validation accuracies for predicting land-use change. The overall cross-validation accuracies of a method were calculated by averaging averaged cross-validation accuracies for changes in LU between 2010 and 2015. Thus, methods that produce the highest overall cross-validation accuracies and the highest averaged final model accuracies by LU type were determined. Furthermore, the distribution of accuracy values across types of LU change was determined.

### 2.5. Implementation of Statistical Approaches

The R statistical software [42] was used to implement the four statistical methods evaluated. Data partitioning for CCV and SCV was done using the *creatFolds* function from the caret package [43] and *partition.kmeans* function from the sperrorest package [40], respectively. The creation of the transition probability matrices for the MC method used the *prop.table* function, which was applied to a contingency table of land-use classes. The fitting of CCV and SCV models for GAM was conducted using the *train* function that adopts the algorithm of GAM from the mgcv package [44–48]. The *train* function with gam from mgcv package fits predictors with default smoothing functions, the thin plate regression splines that are considered robust smoothers regardless of the dimension of basis functions, which cannot be modified. The *coxph* function in the survival package [49] was used to construct a Cox proportional hazard model [37]. Since the *coxph* function in R can only handle right-censored data, all time measurements are considered precise and only right-censored data (i.e., parcels have not experienced land-use change at the end of study) exist in this study.

## 3. Results

### 3.1. Conventional Cross-Validation and Final Models

Results from the 10-fold CCV by method demonstrated that the overall CCV accuracy was highest for GAM, followed by LR, SA, and MC for both the full- and reduced-balanced training datasets (Table 2). GAM achieved 85.17 percent and 82.39 percent overall accuracies for full- and reduced-balanced training datasets, respectively. For full-balanced training datasets, the overall accuracy of GAM is 4.2 percent, 4.26 percent and 38.12 percent higher than the overall accuracies of LR, SA, and MC, respectively. For reduced-balanced training datasets, the overall accuracy of GAM is 2.15 percent, 2.6 percent and 31.71 percent higher than the overall accuracies of LR, SA, and MC, respectively. Excluding MC, the increase in overall CCV accuracy due to reducing the sample size from the full-balanced to the reduced-balanced dataset was highest for GAM (2.78 percent). The effects of samples size were more strongly observed among individual land-use changes. For example, the largest difference among the land-use changes occurred for medium-density residential under GAM (6.98 percent).

Despite the higher computational overhead, GAM outperformed the other three modelling approaches in overall accuracy and yielded the highest CCV accuracy for six of eight land-use changes with full-balanced datasets (medium-density residential, high-density residential, commercial, transportation, protected area and recreation, under development) and five of eight land-use changes with reduced-balanced datasets (high-density residential, commercial, transportation, protected area and recreation, under development; Table 2). While LR obtained the highest accuracy for industrial, the benefit was minor (0.1 percent difference) relative to GAM. In contrast, SA had an accuracy 5.1 percent greater than GAM for low-density residential and had an accuracy 2.18 percent higher than GAM for

the reduced-balanced medium-density residential land use. The MC approach was able to contribute to the three land-use types (institution, agriculture and water) that could not be modelled by the other methods and performed surprisingly well for agriculture due to the observed low frequency of agricultural land-use changes in the data.

**Table 2.** The averaged and overall 10-fold CCV accuracy (percentage) for full-balanced (FB) and reduced-balanced (RB) training datasets. GAM: generalized additive model.

| Method / Land-Use Change | MC FB | MC RB | LR FB | LR RB | GAM FB | GAM RB | SA FB | SA RB |
|---|---|---|---|---|---|---|---|---|
| Low-Density Residential | 35.79 | 46.90 | 68.62 | 68.62 | 65.85 | 65.85 | **70.95** | **70.95** |
| Medium-Density Residential | 67.46 | 30.69 | 89.62 | 88.44 | **93.98** | 87.00 | 89.56 | **89.18** |
| High-Density Residential | 42.79 | 39.19 | 71.10 | 69.88 | **79.46** | **77.71** | 71.64 | 70.26 |
| Commercial | 24.62 | 29.92 | 80.89 | 81.80 | **91.61** | **86.53** | 78.45 | 76.25 |
| Industrial | 45.58 | 50.31 | **90.05** | **90.05** | 89.95 | 89.95 | 88.84 | 88.84 |
| Institution | **60.00** | **80.00** | n/a | n/a | n/a | n/a | n/a | n/a |
| Transportation | 56.12 | 56.08 | 79.68 | 76.15 | **83.08** | **79.72** | 79.76 | 76.71 |
| Protected Area and Recreation | 24.43 | 42.01 | 87.24 | 87.24 | **89.80** | **89.80** | 85.52 | 85.52 |
| Agriculture | **94.83** | **95.74** | n/a | n/a | n/a | n/a | n/a | n/a |
| Water | **56.56** | **57.85** | n/a | n/a | n/a | n/a | n/a | n/a |
| Under Development | 9.33 | 28.78 | 82.93 | 82.13 | **87.66** | **82.56** | 82.54 | 80.57 |
| Overall | 47.05 | 50.68 | 80.97 | 80.24 | **85.17** | **82.39** | 80.91 | 79.79 |

Note: Bold values indicate highest accuracy by land-use type.

Differences in averaged land-use accuracies among methods were greater than differences between overall accuracies (Table 2). Excluding MC, the differences in averaged accuracies across the other three models were lowest for industrial (1.21 percent) and greatest for full-balanced commercial (13.16 percent). These results suggest that model choice is critical to gaining an accurate representation of pattern for some types of land-use change. In addition, our results suggest that greatly increasing the sample size from reduced-balanced to full-balanced for land-use changes of medium-density residential, high-density residential, commercial and under development had little effect on the within land-use accuracy for LR, whereby the difference is less than 1.18 percent for all land-use types except transportation (3.53 percent). The differences between sample sizes for individual types were greater for GAM and SA (Appendix D).

The final GAM, LR, and SA models were those that produced the highest accuracy in the 10-fold CCV process for specific types of land-use change and held significant predictors. The set of final models were evaluated against partitioned test datasets. The ranking of the overall final model accuracies by method was the same as that obtained from CCV (i.e., GAM > LR > SA) with reduced-balanced training data. The overall accuracy of SA was slightly higher than the overall accuracy of LR for final models with full-balanced test data. Since MC does not have a specific form and does not contain any predictor variables, it was excluded from this comparison.

The overall final models' accuracies with full-balanced test datasets are again slightly higher than the overall final models' accuracies with reduced-balanced test datasets. For full-balanced test datasets, the overall accuracy of GAM is 4.37 percent and 4.27 percent higher than the overall accuracies of LR and SA, respectively. For reduced-balanced test datasets, the overall accuracy of GAM is 2.41 percent and 3.13 percent higher than the overall accuracies of LR and SA, respectively. This implies that the advantage of GAM predicting overall land-use changes with full-balanced dataset over reduced-balanced dataset is higher in the final models. Despite the higher accuracy in the final models, the differences between the three models are greater and suggest that the GAM is not only the best overall model, but also that it is more generalizable than LR and SA.

Among the final models GAM had the highest accuracy for six land-use changes (medium-density residential (full-balanced), high-density residential, commercial, industrial, transportation, and under development), SA performed best for two land-use changes (low-density residential and medium-density residential (reduced-balanced)), and LR performed best for two land-use changes

(protected area and recreation, and under development (reduced-balanced), Table 3). Again, the differences among land-use change accuracies were much greater than the difference in overall accuracies. The coefficients of significant predictors in final models are listed in Appendix E.

**Table 3.** The individual and overall accuracy (percentage) for final models derived from 10-fold CCV with full-balanced (FB) and reduced-balanced (RB) test datasets.

| Method Land-Use Change | LR | | GAM | | SA | |
|---|---|---|---|---|---|---|
| | **FB** | **RB** | **FB** | **RB** | **FB** | **RB** |
| Low-Density Residential | 67.05 | 67.05 | 70.11 | 70.11 | **70.50** | **70.50** |
| Medium-Density Residential | 89.83 | 87.63 | **93.78** | 89.30 | 90.01 | 89.90 |
| High-Density Residential | 69.22 | 67.33 | **78.24** | **76.67** | 69.35 | 68.67 |
| Commercial | 78.82 | 83.67 | **92.15** | **90.17** | 80.12 | 81.67 |
| Industrial | 91.19 | 91.19 | **91.82** | **91.82** | 91.19 | 91.19 |
| Institution | n/a | n/a | n/a | n/a | n/a | n/a |
| Transportation | 79.05 | 79.67 | **83.98** | **80.00** | 78.70 | 76.59 |
| Protected and Recreation | **90.32** | **90.32** | 88.39 | 88.39 | 88.24 | 88.24 |
| Agriculture | n/a | n/a | n/a | n/a | n/a | n/a |
| Water | n/a | n/a | n/a | n/a | n/a | n/a |
| Under Development | 83.86 | **82.33** | **85.87** | 82.00 | 82.05 | 76.67 |
| Overall | 81.17 | 81.15 | **85.54** | **83.56** | 81.27 | 80.43 |

Note: Bold values indicate highest accuracy by land-use and land-cover type.

### 3.2. Spatial Cross-Validation and Final Models

To evaluate the effects of spatial autocorrelation on model performance, spatial cross-validation (SCV) was conducted, which produced similar overall rankings to those attained in the CCV for the full-balanced dataset (i.e., GAM > SA > LR > MC). Results of the SCV using the reduced-balanced dataset held slightly different results with LR outperforming SA (i.e., GAM > LR > SA > MC) (Table 4). The overall SCV accuracies were slightly lower than the CCV accuracies (Table 4) and the differences in accuracies among the methods were also slightly reduced.

**Table 4.** The averaged and overall 10-fold SCV accuracy (percentage) for full-balanced (FB) and reduced-balanced (RB) training datasets.

| Method Land-Use Change | MC | | LR | | GAM | | SA | |
|---|---|---|---|---|---|---|---|---|
| | **FB** | **RB** | **FB** | **RB** | **FB** | **RB** | **FB** | **RB** |
| Low-Density Residential | 35.37 | 50.76 | 67.43 | 67.43 | 66.06 | 66.06 | **69.15** | **69.15** |
| Medium-Density Residential | 68.35 | 28.20 | 89.38 | 89.67 | **92.87** | 86.78 | 89.25 | **90.35** |
| High-Density Residential | 41.09 | 40.80 | 66.72 | 69.50 | **73.92** | **78.19** | 68.86 | 72.03 |
| Commercial | 25.41 | 28.36 | 79.82 | 78.92 | **88.29** | 78.59 | 78.99 | 77.70 |
| Industrial | 45.56 | 45.57 | 89.37 | 89.37 | **90.52** | **90.52** | 90.03 | 90.03 |
| Institution | **80.00** | **80.00** | n/a | n/a | n/a | n/a | n/a | n/a |
| Transportation | 57.82 | 60.32 | 76.77 | 76.91 | **82.52** | **78.80** | 79.40 | 76.54 |
| Protected Area and Recreation | 27.78 | 45.57 | **85.68** | **85.68** | 84.51 | 84.51 | 84.32 | 84.32 |
| Agriculture | **90.71** | **97.36** | n/a | n/a | n/a | n/a | n/a | n/a |
| Water | **63.78** | **41.65** | n/a | n/a | n/a | n/a | n/a | n/a |
| Under Development | 9.90 | 27.29 | 77.09 | **78.98** | 78.58 | 78.92 | 74.98 | 76.34 |
| Overall | 49.62 | 49.63 | 79.03 | 79.56 | **82.16** | **80.30** | 79.37 | 79.56 |

Note: Bold values indicate highest accuracy by land-use type.

The differences in overall SCV accuracies due to sample size within methods were minor (within 2 percent). Furthermore, the SCV results do not uniformly demonstrate that sample size positively influences overall SCV accuracies, which is likely due to the heterogeneity in the reduced-balanced dataset introduced due to partitioning the training and testing data spatially. By partitioning the training and testing data spatially, models are constructed that have reduced variation in predictor values but the differences among the derived models are greater. The outcome of averaging these

differences yields results that are more dependent on the spatial segmentation, which differs from the complete and random mixing of partitions generated from the CCV. The greater mixing in CCV yields more similar models and a clear effect of sample size on model performance.

In contrast, sample size had a substantial effect on some SCV specific land-use change accuracies (e.g., GAM medium-density residential had >6% difference between full- and reduced-balanced). In general, GAM performed better using the full-balanced dataset, producing the highest accuracy models for six land-use changes (medium-density residential, high-density residential, commercial, industrial, transportation, and under development) and only the highest accuracy model for three land-use changes with reduced-balanced training dataset (high-density residential, industrial, and transportation). LR performed best for commercial, protected area and recreation, and under development land-use changes with the reduced-balanced training dataset and SA performed best for low-density residential regardless of sample size, and medium-density residential with reduced-balanced training dataset. Similar to the CCV, using SCV, MC was able to contribute to those land-use changes that could not be modeled using the other approaches (institution, agriculture and water). The SCV results corroborate the CCV results in that model choice is critical to gaining an accurate representation of pattern for some land-use change types.

Among the final models derived from SCV, GAM performed the best overall as well as for the following full-balanced dataset land-use changes: medium-density residential, high-density residential, commercial, transportation and under development (Table 5). LR produced the second highest overall final model accuracies and best models of land-use change for low-density residential, industrial, protected area and recreation, for both full- and reduced-balanced datasets and tied SA for the highest under development accuracy using the reduced-balanced dataset. These results demonstrate that the model accuracies ranked with GAM > LR > SA, which is the same outcome as the SCV reduced-balanced dataset results. Results among specific land-use types and sample sizes confirm the findings made by analyzing results from CCV and CCV-final models that model choice is a more critical factor than sample size is for making predictions of land-use changes. Furthermore, SCV only contributed to alleviating over-fitting by a small amount as demonstrated by the minor reduction in overall model accuracies. We conclude that spatial autocorrelation in our data did not cause substantial over-fitting of our statistical models in terms of their overall accuracies but did have a recognizable impact on individual land-use changes and single methods. The coefficients of significant predictors in final models are listed in Appendix E.

**Table 5.** The individual and overall accuracy (percentage) for final models derived from 10-fold SCV with full-balanced (FB) and reduced-balanced (RB) test datasets.

| Method<br>Land-Use Change | LR | | GAM | | SA | |
|---|---|---|---|---|---|---|
| | **FB** | **RB** | **FB** | **RB** | **FB** | **RB** |
| Low-Density Residential | **70.88** | **70.88** | 69.35 | 69.35 | 67.82 | 67.82 |
| Medium-Density Residential | 88.20 | 89.67 | **93.71** | **92.98** | 89.89 | 90.00 |
| High-Density Residential | 69.61 | 67.33 | **77.65** | **79.33** | 70.00 | 67.33 |
| Commercial | 81.33 | 80.27 | **91.37** | **82.94** | 80.62 | 79.67 |
| Industrial | **97.48** | **97.48** | 94.34 | 94.34 | 91.19 | 91.19 |
| Institution | n/a | n/a | n/a | n/a | n/a | n/a |
| Transportation | 79.58 | 77.00 | **83.98** | **83.33** | 78.70 | 77.00 |
| Protected Area and Recreation | **89.30** | **89.30** | 87.74 | 87.74 | 87.74 | 87.74 |
| Agriculture | n/a | n/a | n/a | n/a | n/a | n/a |
| Water | n/a | n/a | n/a | n/a | n/a | n/a |
| Under Development | 82.96 | **82.33** | 84.75 | 81.67 | 81.61 | **82.33** |
| Overall | 82.42 | 81.78 | **85.36** | **83.96** | 80.95 | 80.39 |

Note: Bold values indicate highest accuracy by land-use type.

### 3.3. The Combination of Statistical Methods

A theoretical land-use model was constructed by combining methods that produced the highest final model accuracies by land-use type (Table 6). Since SCV-final models generally alleviated over-fitting caused by spatial autocorrelation, final models derived from SCV were selected to form the theoretical method. Moreover, MC derived from CCV was selected since MC only accounted for frequencies of land-use changes.

**Table 6.** The combination of final models that produces the highest accuracy (percentage) by land-use type with full-balanced (FB) and reduced-balanced (RB) test datasets.

| Land-Use Change | FB | | RB | |
|---|---|---|---|---|
| | Method | Accuracy | Method | Accuracy |
| Low-Density Residential | LR-SCV | 70.88 | LR-SCV | 70.88 |
| Medium-Density Residential | GAM-SCV | 93.71 | GAM-SCV | 92.98 |
| High-Density Residential | GAM-SCV | 77.65 | GAM-SCV | 79.33 |
| Commercial | GAM-SCV | 91.37 | GAM-SCV | 82.94 |
| Industrial | LR-SCV | 97.48 | LR-SCV | 97.48 |
| Institution | MC-SCV | 25 | MC-SCV | 40 |
| Transportation | GAM-SCV | 83.98 | GAM-SCV | 83.33 |
| Protected Area and Recreation | LR-SCV | 89.30 | GAM-SCV | 89.30 |
| Agriculture | MC-SCV | 96.58 | MC-SCV | 96.03 |
| Water | MC-CCV | 55 | MC-SCV | 53.85 |
| Under Development | GAM-SCV | 84.75 | LR-SCV/SA-SCV | 82.33 |
| Overall [1] | | 78.70 | | 78.95 |
| Overall [2] | | 86.14 | | 84.82 |

Note: Overall [1] is the overall accuracy of all land-use changes. Overall [2] is the overall accuracy excluding land-use changes of institution, agriculture and water.

The theoretical land-use model produces an overall accuracy that is 3.98 percent and 4.52 percent higher than the best performing GAM models derived from SCV with full- and reduced-balanced datasets, respectively. When institution, agriculture and water are included (as represented by the MC model) the overall accuracy drops from 86.14 percent to 78.70 percent for full-balanced datasets and from 84.82 percent to 78.95 percent for reduced-balanced datasets; however, it is only through this mixed approach that all types of land-use changes can be modeled.

### 4. Discussion

The presented research sought to answer the question: what is the overall accuracy of different statistical methods in representing land-use change. To answer this question we developed, quantified, and compared the accuracy of four different conceptual approaches to modelling land-use change, i.e., parametric model, non-parametric model, time series, and stochastic process as represented by logistic regression (LR), general additive modelling (GAM), survival analysis (SA), and Markov chain (MC), respectively. These approaches were used to generate models of eleven different land-use types and were tested using 2010 and 2015 data for the Region of Waterloo, Ontario, Canada. Our results found that overall model accuracy was highest for GAM followed by SA, LR, and MC using CCV and final model selection held that ranking. When SCV was conducted, LR outperformed SA and MC was not competitive in either case. Given the close performance of GAM, LR, and SA, it is worth noting that the run time for the 10-fold CCV for LR, SA, and MC were less than 20 s (Appendix D). However, performing the 10-fold CCV for GAM took over eighty-five minutes for both full- and reduced-balanced datasets due to the iteration in back-fitting of smoothing functions.

We subsequently sought to answer the question: how do the accuracies differ among the tested statistical methods for specific land-use types? While GAM, again, outperformed the other modelling approaches, some specific land-use types were better modeled using each of LR, SA, and MC. In fact, the differences among methods by land-use type were greater than the overall model accuracies by method. To interrogate this concept further, all model derivations and analyses were repeated with two

differently sized datasets, full-balanced and reduced- balanced, and a theoretical model comprising the best performing models of each type was derived. Our results demonstrate that not only is the choice of model by land-use type more important than sample size, but also that a hybrid land-use model comprising the best statistical modelling approaches for each land-use change outperformed the individual statistical approaches.

It is worth noting that while we found accuracies to typically increase with sample size, this was not the case when SCV was used in model derivation and selection. The inclusion of SCV slightly reduced the degree of over-fitting of models using CCV and demonstrated that spatial autocorrelation had little effect on our final model accuracies. It is also worth noting that while MC was not competitive, it was useful in providing representation of land-use changes that had small sample sizes and could not be represented using other methods or in other cases where there is no predictor data. Given that it has been shown to outperform other null models [50] and performs better when there are data over longer time scales [51,52], it should not be completely disregarded.

### 4.1. Modelling One-to-One Land-use Change versus Many-to-One Land-Use Change

LR and GAM that have been used to model land-use change are usually designed to represent one-to-one land-use change (e.g., with only two classes comprising non-urban to urban [15,53]; non-forest to forest and forest to non-forest [54]). SA has been also been used to model one-to-one land-use change using two land-use classes, non-urban to urban [15], and three one-to-one models of land-use changes from agriculture to three types of subdivisions [6]. In this way, focusing on modelling one-to-one land-use change can reveal the effects of predictors on a specific transition between land uses providing more insight toward potential causal mechanisms for change.

Modelling specific land-use change transitions (i.e., one-to-one) is not only time-consuming when many land-use classes are involved, but the availability of land-use change samples can also be restrictive. For example, if our eleven land-use classes ($n = 11$) were modeled based on single transitions (one-to-one; $r = 2$ land uses; Figure 3a) then we could have ($n!/(n − r)!$) or 110 models. While some of these transitions would be unlikely (e.g., high-density residential to agriculture), the number of samples of many different individual transitions is small for our study area. In contrast, we represented each unique land use as the endpoint resulting from many possible points of origin (Figure 3b), yielding only eleven final models and overall accuracies greater than 85% for our best final models and theoretically combined model with full-balanced datasets.

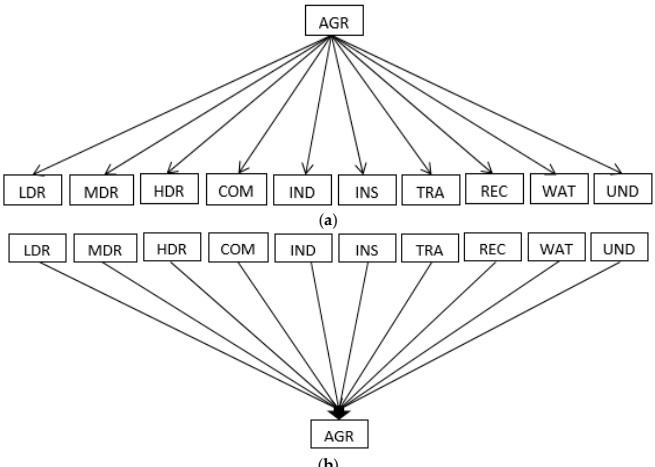

**Figure 3.** (**a**) The ten possibilities of modelling one-to-one land-use change from agriculture to all other land uses. (**b**) Modelling many-to-one land-use change from any land use to agriculture. [Notes: LDR: low-density residential; MDR: medium-density residential; HDR: high-density residential; COM: commercial; IND: industrial; INS: institution; TRA: transportation; REC: protected area and recreation; AGR: agriculture; WAT: water; UND: under development].

*4.2. Operationalizing the Combined Statistical Model*

Prediction of future land uses in an area involves land-use competition, which was not modeled in the presented research. However, the statistical methods were chosen because they produce similar outputs in terms of probability of change to different land-use types. Future research will seek to determine the interactive effects between the tested modelling approaches and evaluate different decision-making algorithms for determining future land use at a given location. Existing models incorporate preferential ordering (e.g., [55]), highest fitness (e.g., [56]), and rescaled roulette-wheel approaches (e.g., [52]) among others. To the best of the authors' knowledge, a comparison among competing land-use models for land-use allocation to a specific location has not been published. The conditions of their success, requirements for application, and lessons learned would not only guide land-use change modelling researchers but would also help to synthesize what is typically a set of disparate case study models that have yet to coordinate the creation of community frameworks, models, or model components [13,57].

*4.3. Future Directions*

The presented comparative analysis focused on four methods that are most frequently used in land-use modelling literature and span four conceptual approaches to statistical analysis (stochastic process, parametric model, non-parametric model, and time series model). A variety of other approaches would be worth investigating, such as mixed effects models (e.g., [58]), especially mixed effects logistic regression and mixed effects additive logistic regression as well as machine learning techniques such as random forest (e.g., [59]) and support vector machine (e.g., [60]). Mixed effects logistic regression and mixed effects additive logistic regression are expected to perform better than LR and additive logistic regression, respectively, since random effects in mixed-effects models could take account of time-varying variables and spatial autocorrelation, but at the cost of increased model complexity and difficulty of interpretation. Machine learning approaches have been applied to classify land cover (e.g., [61–64]) and are more flexible than traditional statistical methods since they are spared from general assumptions of statistical methods such as linearity, independence, and an underlying distribution of data. Moreover, machine learning can benefit from a large amount of input data, both observations and predictors, and is less affected by multi-collinearity among predictors. We intend to extend and compare the presented research with these additional methods in the future and welcome others to do so as well using our data included with this publication.

Our comparison of statistical methods used traditional cross-validation and accuracy assessments. The inclusion of spatial cross-validation mitigated the effects of spatial autocorrelation and violation of independence among samples, which yielded models with slightly greater error than those derived from traditional cross-validation. These efforts differ from those used in land-use modelling that focus on partitioning quantity and location of change based on generated maps compared to observations as done by [7]. We created balanced datasets that comprise locations of change for a specific land use and locations where that type of land use persisted throughout the time frame of our analysis. The tested models were constrained to these data and were not applied against all potential locations as is done in most models of land-use change. Using our approach we did not partition our results by quantity of change and location of change as the quantity remained fixed by the size of our balanced datasets. Future research will assess the maps generated from the tested approaches along with additional approaches such as mixed effects and machine learning as mentioned above.

In making comparisons between the presented model and other models of land-use change, it is worth noting that the presented models used vector parcel data as the minimum mapping unit rather than raster cells. The spatial scale was fixed to the spatial unit at which land-use change decisions are made (i.e., the ownership property parcel). Like raster-based land-use models, our generated maps will experience an increase in accuracy if aggregated to more coarse resolutions, but these have less conceptual meaning given our ownership property parcel approach. While use of property parcels offers alignment with observations that is more realistic than results obtained from raster-based

models, modelling future scenarios using our approach faces the challenge of representing the process of parcelization or property parcel subdivision. Efforts in parcelization are underway (e.g., [65]), however, it remains an area in need of continued research.

We have shown the capacity for statistical models to represent land-use change with relatively high levels of accuracy, however, they are limited in their ability to represent the conditions, actors, behaviors, and decision-making processes that drive land-use change. To represent heterogeneity within the environment, actors driving change within the land system, and their interactions with each other and their environment, agent-based modelling provides an alternative to the constraints of statistical modelling. While agent-based modelling is widely used to represent land-use change [66], the approach faces challenges in the time required for model development, its ability to predict future outcomes, and a lack of available data about actor characteristics, behavior, and interactions [67]. Furthermore, our understanding of statistical models has history and a level of transparency that makes their communication in publications more feasible and subsequently their adoption and use by other non-creators more likely [13].

As the study of complexity complements reductionism rather than displaces it [68], so too can statistical models form the basis for agent-based approaches and the creation of hybrid models that utilize the best aspects of both approaches can be developed. For example, in the absence of data about actors driving industrial and commercial development, logistic regression was used to create a model rich in the number of land-use types (11) [69,70]. Not only did the logistic regression perform well, but few researchers are engaged in modelling the site-selection behavior of industrial and commercial actors using an agent-based modelling approach. A unique and related result from our statistical model comparison is that the land-use change that typically yielded the lowest accuracy across GAM, LR, and SA approaches was low-density residential. This outcome provides justification for the substantial number of agent-based models developed to represent urban sprawl and residential development (e.g., [71–73]), which could be argued as the most prominent focus among case study applications of agent-based models to land-use change in developed countries, with farming agents yielding a close second.

Current effort in the land-use modeling community using agent-based modelling approaches involve attempts to advance toward representing land-use change across large spatial extents [74]. However, the challenges associated with deriving common agent types (e.g., human function types [14]) and collecting actor characteristic, behavior, and decision-making data are inhibiting these efforts. One approach to simplify the process and speed up advancements in this area is to construct hybrid models that use an agent-based framework and statistical models of land-use change as placeholders for agent behavioral models until novel approaches to data collection or agent behavioral development take place. The incorporation of statistical representations of agent behaviors can (1) produce outcomes repeatedly in a relatively short time period, (2) can be validated, and (3) can be applied across large spatial extents. With a hybrid ABM-statistical approach, one can get a representative model up and running quickly, and can provide a range of insights and findings that can be extended when behavior data become available. Such an approach may also facilitate the representation of human systems with natural systems, which has faced not only technical challenges, but also buy-in from natural scientists and funding agencies [75].

## 5. Conclusions

Parametric, non-parametric, time series, and stochastic models of land-use change were created and compared across three dimensions: accuracy (overall and by land-use type), sample size, and spatial independence via conventional versus spatial cross validation. To the best of the authors' knowledge, this is the first study to formally compare the relative performance of these statistical approaches as represented by LR, GAM, SA, and MC models of land-use change at the same location using the same data. Our results indicated that GAM outperformed the other three approaches in both overall accuracy and among most land-use accuracies. However, LR and SA were more accurate

for specific land-use types (e.g., low-density residential and industrial) and MC was able to represent changes that could not be modeled by other approaches due to sample size restrictions. Therefore, a land-use model that combines the best models of each land-use type can outperform any one modelling approach and provide a richer representation of land use types.

The performance of the four tested methods and their relative ranking may differ for other regions since they are dependent on the quality and characteristics of the data to which they are applied and the underlying processes of land-use change may differ among regions. While we assessed the fit of statistical models to the outcomes of land-use change processes, a better fit may not improve our explanation and understanding of these processes or the models predictive capacity due to over-fitting. Despite these caveats, the presented comparison, of statistical land-use models, offers guidance to others selecting among statistical approaches for representing land-use change. Furthermore, the effects of sample size on model performance were quantified and were minor relative to model choice for specific types of land-use change, particularly when spatial cross-validation is incorporated into the modelling process. While the incorporation of spatial cross-validation reduced overall model performance, it also reduced over-fitting and illustrated that spatial autocorrelation had little effect on our modelling analysis.

The presented research contributes to a lack of literature on land-use change modelling where the focus is on high-fidelity parcel-level models, which is the scale at which land-use change decisions are made [69]. Future research will seek to extend the modelling comparison to other statistical modelling approaches as well as to investigate the outcomes of different scenarios of future land-use and land-cover change. We hope that by providing the details about the results in the paper and supplementary material about the model specifics in appendices that others will compare and contrast their approaches to the work presented here.

**Supplementary Materials:** The following are available online at http://www.mdpi.com/2073-445X/7/4/144/s1.

**Author Contributions:** conceptualization, D.T.R. and B.S.; methodology, B.S.; validation, B.S.; formal analysis, B.S.; investigation, B.S.; resources, D.T.R.; data curation, B.S.; writing—original draft preparation, B.S.; writing—review and editing, D.T.R.; visualization, B.S.; supervision, D.T.R.; project administration, D.T.R.; funding acquisition, D.T.R.

**Funding:** This research was funded by the Natural Sciences and Engineering Research Council (NSERC, Ottawa, ON, Canada) of Canada under Discovery Grant (06252-2014).

**Acknowledgments:** The authors would like to thank Alex Smith for his efforts in creating the land-use data and collaboration on our manual land-use classification protocol. We would also like to thank the anonymous reviewers and the editor for their insightful feedback, which has improved the quality of the presented research.

**Conflicts of Interest:** The authors declare no conflict of interest.

## Appendix A

The need of a manual classification arose when misclassification found in computer simulated 2010's land-use data created by Alexander Smith in 2016. The manual classification of 2010's land use was conducted for the Region of Waterloo with the support of 2010's SWOOP imagery. The rules of manual land-use classification were the result of discussion between Smith and the author, which can be found in [22].

After the manual classification, three small areas in the Region of Waterloo were randomly chosen and compared with the classification result from computer simulated land-use data. The result of comparison has achieved an overall accuracy of 90 percent. Later, an overall accuracy of the computer simulated 2010 land-use data in the whole study area was computed using manually classified land-use data as reference, which is about 88 percent. Thus, the classification of computer simulated 2010 land-use data was considered satisfactory. Similarly, computer-simulated 2006 and 2015 land-use data were also considered satisfactory since all land-use data were classified using the same rules, methods and technology. Therefore, computer simulated land-use data of 2006, 2010,

and 2015 were used as ground truth data toward modelling land-use changes with the proposed statistical methods.

Before creating any variables, DAUIDs (i.e., unique IDs of Das) in the Region of Waterloo in 2010 were assigned to parcels according to the location of parcels in the DA in order to relate DA's information to parcels. Geometries (i.e., perimeter and area) of the parcel polygons and DA polygons were calculated in ArcGIS and attached to the ownership parcel data. Site variables (i.e., mean slope and mean elevation) were created using slope and elevation data. Demographic data from 2011 Census data (e.g., population) associated with the DA in the Region of Waterloo retrieved from Statistics Canada were merged with the parcel data in excel format in R by the common variable DAUID. The parcel data were then imported back to ArcGIS since some spatial variables need to be created using the Polygon Neighbors tool in ArcGIS. The tool created a table that contains IDs of source polygons, IDs of neighbor polygons and records of land-use types for neighbor polygons. The table was then used to calculate the percentage of each land-use type around each source polygon by manually programmed codes in R. The percentage of neighbor land use was merged with the parcel data by unique parcel ID (i.e., ID of source polygon) in R. Furthermore, the parcel data were converted to point data (i.e., centroids of parcels) in order to calculate Euclidean distances from parcel centroids to other features (e.g., highway ramp and commercial parcel) in ArcGIS. During the process of variable creation, some parcels have been removed from the full dataset due to the lack of data for creating drivers associated with the parcels (e.g., a lack of census data in some areas).

Misclassification was found in some rare cases of land-use change during the process of creating status for land-use change (i.e., binary response variables). Since rare cases of land-use change (e.g., from industrial to water) are a relatively small amount of data compared to the total, second round of manual classification was conducted by the author alone to increase the classification accuracy of the data. During this round, SWOOP imageries for all three years (i.e., 2006, 2010, and 2015) were used to classify land-use types that associated with parcels being found with the occurrence of rare cases of land-use change. The manual classification was also done to parcels that found with the occurrence of some other land-use changes that have a relatively small amount of data compared to the total due to the consideration of data accuracy. Meanwhile, some rules of manual land-use classification have been modified based on Smith's work. The complete and modified rules are presented in Appendix C. Moreover, spatial variables were re-created using 2010 as the reference year since the neighborhood parcels may be changed.

Furthermore, some other problems have arisen. As mentioned in Section 2.4 in this paper, each full dataset is consisted of all parcels that had their land use converted to a specific land use in 2015 and parcels that had remained the specific land use during the study period. After conducting an explanatory analysis on full datasets, it has been found that the majority types of land-use change in a full dataset came from the part of parcels that has not been manually verified or modified in the second round. Therefore, it is reasonably to suspect the accuracy of the data. However, the study has moved forward with the current set of data due to the high overall accuracy of computer simulated land-use data and the time constraint of this project. Another finding is that the classification accuracy of computer simulated 2010 land use is about 68 percent for the approximately 6000 parcels that have been verified or modified in the second round of manual classification. This infers that the computer classification approach performed differently for different land-use types. In addition, the season that SWOOP data have been taken is another factor that can cause misclassification of the same land use in different years by computer simulation methods.

## Appendix B

**Table A1.** Names and Description of Predictors.

| Name | Description (Unit) |
|---|---|
| lu2015 | 2015 land-use of a parcel in the Region of Waterloo |
| lu2010 | 2010 land-use of a parcel in the Region of Waterloo |
| lu2006 | 2006 land-use of a parcel in the Region of Waterloo |
| lc2010 | 2010 land-cover of a parcel in the Region of Waterloo |
| lc2006 | 2006 land-cover of a parcel in the Region of Waterloo |
| DA_Area | Area of a DA ($km^2$) |
| Parcel_Area | Area of a parcel ($km^2$) |
| MeanDEM | Mean elevation of a parcel (km above sea level) |
| MeanSlope | Mean slope of a parcel (degree) |
| Ramp_dist | Distance from the centroid of a parcel to the nearest highway ramp (km) |
| River_dist | Distance from the centroid of a parcel to the nearest river (km) |
| Water_dist | Distance from the centroid of a parcel to the nearest water body (km) |
| Wood_dist | Distance from the centroid of a parcel nearest wooded area (km) |
| Lroad_dist | Distance from the centroid of a parcel to the nearest local road (km) |
| Mroad_dist | Distance from the centroid of a parcel to the nearest main road (km) |
| lu4_dist | Distance from the centroid of a parcel to the nearest commercial parcel (km) |
| lu5_dist | Distance from the centroid of a parcel to the nearest industrial parcel (km) |
| lu8_dist | Distance from the centroid of a parcel to the nearest protected area/recreational parcel (km) |
| lu9_dist | Distance from the centroid of a parcel to the nearest agricultural parcel (km) |
| DA_population_density | Population density in a DA (population in DA/DA area) in 2010 (person/$km^2$) |
| Residential_population_density | Residential population density in a DA (population in DA/total residential areas in DA) in 2010 (person/$km^2$) |
| Change_Population | The rate of change of population from 2006 to 2011 based on the DA a parcel resides |
| Change_AveIncome | The rate of change of average income from 2006 to 2011 based on the DA a parcel resides |
| F_lu1 | Proportion of low-density residential parcels around a parcel |
| F_lu2 | Proportion of median-density residential parcels around a parcel |
| F_lu3 | Proportion of high-density residential parcels around a parcel |
| F_lu4 | Proportion of commercial parcels around a parcel |
| F_lu5 | Proportion of industrial parcels around a parcel |
| F_lu6 | Proportion of institution parcels around a parcel |
| F_lu7 | Proportion of transportation parcels around a parcel |
| F_lu8 | Proportion of protected area/recreation parcels around a parcel |
| F_lu9 | Proportion of agricultural parcels around a parcel |
| F_lu10 | Proportion of water parcels around a parcel |
| F_lu11 | Proportion of developing parcels around a parcel |

Note: Variables listed in this table are the variables actually being used to construct models in this study. Some variables have been created were excluded from model building since they are highly correlated with some variables listed in this table.

## Appendix C

The original tables can be found in the Appendix section in Smith (2017).

**Table A2.** Manual land-use classification of parcels in the Region of Waterloo.

| # | Name | Classification Description Based on Perceived Uses and Services |
|---|---|---|
| 1 | Low-Density Residential | Parcels that appear to contain a single dwelling for a single family on a large property. These parcels typically appear outside the urban core in suburbs or rural areas. While houses tend to be larger than medium density residential, it is not a requirement for the classification. |
| 2 | Medium-Density Residential | Average sized parcels containing a single dwelling for a single family, which may or may not be attached to adjacent dwellings. This class contains the majority of residential parcels within subdivisions and the urban core. In most parcels, the house and driveway cover most or all of the width of the parcels, with yards in the front and back. Townhouses are usually classified as medium-density residential. * |
| 3 | High-Density Residential | Parcels containing buildings with multiple dwellings or units, and therefore multiple families within the parcel. Typically in two forms, apartment or condo buildings, and townhouses where one parcel contains multiple units. Parcels may contain green space and parking lots in addition to the buildings. |
| 4 | Commercial | Parcels containing business where customers visit to obtain products and services, or office buildings which may not receive customers. Larger parcels, such as malls or box stores, will contain large parking lots for customers. These parcels do not contain large outdoor storage areas, although garden and home improvement stores may have some outdoor storage. |
| 5 | Industrial | Parcels which contain a business with an outdoor storage area such as a factory or a car scrapyard. These business typically do not receive customers although there may be parking lots for employees and areas for incoming materials and outgoing products. |
| 6 | Institutional | Manually classified parcels for schools (private and public) and hospitals. Schools and hospitals can appear as a variety of classes but provide different services from these misclassifications (e.g., Commercial or Protected Areas and Recreation). Manually classifying these parcels allows for them to be included in the landscape without large amounts of misclassification. |
| 7 | Transportation | Parcels which represent roads and railways. These parcels often include the boulevard and sidewalks. Highway interchange parcels include all the land which is owned and managed by the managing government. |
| 8 | Protected Areas and Recreation | Areas which have a primary purpose of recreation, such as parks, or protected areas such as forests. Commercial forests and private forests are included in this class as they appear very similar, or even identical to the natural forests. |
| 9 | Agriculture | Parcels which are primarily used for raw food production. This includes fields for crops and pastures. Some parcels will have barns and/or a farm house, while others may have neither. Parcels may also include a portion which is forested, sometimes referred to as "the back forty". |
| 10 | Water | Parcels which have a main purpose of outlining waterbodies such as rivers. Lakes are included when the lake occupies a majority of the parcel. The rest of the parcel may include sections which would otherwise be classified as Protected Areas and Recreation. |
| 11 | Under Development | Properties where construction has not been completed and no residents or business has moved in. These parcels may become many different classes when complete, but the class cannot be guaranteed at the time of the imagery. Depending on the progress of a development project, residential areas and big box stores or shopping complexes may appear similar as the area is represented by only a single parcel. |

* Addition for clarification of manual land-use classification used by Smith (2017).

**Table A3.** Clarifications between similar land use classes.

| First Class | Second Class | Problem | Solution |
|---|---|---|---|
| Low-Density Residential | Medium-Density Residential | Parcel size is a continuous variable and it is difficult to define the exact separation between the two classes. | In many cases where there is confusion, the house is the same size as the surrounding properties which are either low or medium density and is a similar distance from the road. The parcel in question will usually have its additional size added through its backyard. If the backyard visually occupies two thirds of the property, it can be easily called low density, if less, medium density. If the parcel has a backyard smaller than two thirds, but the front yard and house are large, then it can also be classified as low density. If an absolute value of size is needed, 2000 m$^2$ should be used as the minimum size for Low Density Residential. |
| Low-Density Residential | Protected Area and Recreation | Household in a large parcel is surrounded by forest or green land with no appearance of backyard/garden. Sometimes the parcel could contain a small portion of backyard/garden relative to the total of the parcel. * | Even the size of the house and the maintained portion of the property is very small compared to the area of the forest, the parcel should be classified as low-density residential. * |
| Medium-Density Residential | Under Development | A house is visible in the parcel that is under development | If there is a completed house with grass on the property it should be considered complete and classified as Medium Density Residential. If the house does not appear complete or there is no grass where there should be, it should be classified as Under Development. |

* Addition for clarification of manual land-use classification used by Smith (2017).

**Table A4.** Exemptions and special cases in land-use classification.

| Example | Class | Reasoning |
|---|---|---|
| Airport | Commercial | Airports provide services similar to Commercial parcels, where people are constantly visiting the parcel. Visually they are similar as they both include large paved areas such as parking lots and a large building. |
| Fire station | Commercial | Although functionally different from Commercial parcels, they are very similar in the imagery. |
| Graveyard | Protected Areas and Recreation | Graveyards and cemeteries are visually similar to parks, where there are paths for people to walk and grass fields. The only visual difference is that there are pieces of stone (headstones) scattered across the fields and there is no sports equipment. |
| Water Tower | Protected Areas and Recreation | Water towers can be visually similar to parks as they can have large grassy areas surrounding the tower. If the water tower is in a parcel without much grassed area, it may be classified as Commercial instead. |
| Commercial Forest –Post-Harvest | Various | If the harvested forest appears to be converted into agriculture, classify as Agriculture. If it shows signs of urban development, it should be classified as Under Development. If it appears to be replanted and is still being used as a commercial forest, classify as Protected Areas and Recreation. |
| Catwalk | Transportation | The paths between houses, or catwalks, are similar to roads, although a little smaller. A path through a park or green space would not be considered transportation. |
| Walking paths | Protected Areas and Recreation | Walking paths in the area can often be found under large electrical transmission lines. The transmission lines and towers account for a small portion of the parcel, and therefore simply appear as grassy corridors through subdivisions, similar to parks. |
| Church | Commercial | Churches are visibly similar to Commercial parcels because they are a building which has a parking lot and some property. Functionally they are also similar as people will visit a church for a relatively short period of time, similar to a business. |
| Artifacts | N/A | The parcel data is not perfect and has artifacts from either previous versions, or mistakes during creation. Some artifacts have little impact on the data, while others have large impacts. The most frequent example is a single parcel being divided into multiple parcels by the artifacts. |
| Artifacts–Splits | N/A | When a parcel is divided by artifacts all segments should be classified as the original type if suitable. If a segment can clearly be classified as another land use type it should be done. For example, if a Low Density Residential parcel is divided into three pieces, two covering the house and one covering a forest at the back of the property, the two on the house should be Low Density Residential and the one on the forest should be Protected Areas and Recreation. |
| Artifacts–Slivers | N/A | Another form of artifact is a sliver. These sliver parcels are very thin and long. Examples can be a few centimeters wide but almost a kilometer long. Sliver parcels should be ignored and not classified if noticed. |
| Mixed Parcels | N/A | Occasionally parcels will contain multiple land use types other than the previously mentioned scenarios. For example a parcel may contain a house and land on one half and part of a waterbody on the other half. In these scenarios where there is no clear majority of land use type the following order of priority should be used: Medium Density Residential > High Density Residential > Low Density Residential > Commercial > Industrial > Institution > Transportation > Under Development > Agriculture > Protected Areas and Recreation > Water |
| Future Development | N/A | In the scenarios where parcels have been created but no development has begun, classify the parcel based on the currently present land use type. If the imagery shows evidence of development, then classify as Under Development. |

## Appendix D

**Table A5.** Running time (seconds) of methods with 10-fold CCV by land-use type for full-balanced (FB) and reduced-balanced (RB) training datasets.

| Method | MC | | LR | | GAM | | SA | |
|---|---|---|---|---|---|---|---|---|
| **Land-Use Change** | **FB** | **RB** | **FB** | **RB** | **FB** | **RB** | **FB** | **RB** |
| Low-density residential | n/a | n/a | 2.93 | 2.93 | 152.06 | 152.06 | 1.14 | 1.14 |
| Medium-density residential | n/a | n/a | 6.21 | 3.56 | 2468.95 | 2816.7 | 4.56 | 1.24 |
| High-density residential | n/a | n/a | 2.92 | 1.02 | 282.7 | 199.04 | 1.41 | 1.2 |
| Commercial | n/a | n/a | 1.55 | 1.03 | 1127.83 | 3408.59 | 2 | 1.27 |
| Industrial | n/a | n/a | 0.83 | 0.83 | 5 | 5 | 1.16 | 1.16 |
| Institution | n/a | n/a | n/a | n/a | n/a | n/a | n/a | n/a |
| Transportation | n/a | n/a | 1.07 | 0.98 | 311.44 | 229.72 | 1.5 | 1.25 |
| Protected area and recreation | n/a | n/a | 0.89 | 0.89 | 17.28 | 17.28 | 1.14 | 1.14 |
| Agriculture | n/a | n/a | n/a | n/a | n/a | n/a | n/a | n/a |
| Water | n/a | n/a | 0.65 | 0.65 | 4.42 | 4.42 | 0.79 | 0.79 |
| Under development | n/a | n/a | 1.06 | 1.03 | 779.71 | 434.11 | 1.47 | 1.24 |
| Total | 17.42 | 8.7 | 18.11 | 12.92 | 5149.39 | 7266.92 | 15.17 | 10.43 |

**Table A6.** Absolute difference between averaged 10-fold CCV accuracies and overall accuracies of full-balanced and reduced-balanced training datasets in percentage (%).

| Method | | | | |
|---|---|---|---|---|
| **Land-Use Change** | **MC** | **LR** | **GAM** | **SA** |
| Low-density residential | 11.11 | 0 | 0 | 0 |
| Medium-density residential | 36.77 | 1.18 | 5.04 | 0.38 |
| High-density residential | 3.60 | 1.22 | 6.98 | 1.38 |
| Commercial | 5.30 | 0.91 | 1.75 | 2.20 |
| Industrial | 4.73 | 0 | 0 | 0 |
| Institution | 20.00 | n/a | n/a | n/a |
| Transportation | 0.04 | 3.53 | 3.36 | 3.05 |
| Protected area and recreation | 17.58 | 0 | 0 | 0 |
| Agriculture | 0.91 | n/a | n/a | n/a |
| Water | 1.29 | n/a | n/a | n/a |
| Under development | 19.45 | 0.80 | 5.10 | 1.97 |
| Overall | 3.63 | 0.73 | 2.78 | 1.12 |

**Table A7.** Absolute difference between overall accuracies of final models derived from 10-fold CCV with full-balanced and reduced-balanced test datasets in percentage (%).

| Method | | | |
|---|---|---|---|
| **Land-Use Change** | **LR** | **GAM** | **SA** |
| Low-density residential | 0 | 0 | 0 |
| Medium-density residential | 2.20 | 4.48 | 0.11 |
| High-density residential | 1.89 | 1.57 | 0.68 |
| Commercial | 4.85 | 1.98 | 1.55 |
| Industrial | 0 | 0 | 0 |
| Institution | n/a | n/a | n/a |
| Transportation | 2.83 | 3.98 | 2.11 |
| Protected area and recreation | 0 | 0 | 0 |
| Agriculture | n/a | n/a | n/a |
| Water | n/a | n/a | n/a |
| Under development | 1.53 | 3.87 | 5.38 |
| Overall | 0.02 | 1.98 | 0.84 |

**Table A8.** Absolute difference between averaged 10-fold SCV accuracies and overall accuracies of full-balanced and reduced-balanced training datasets in percentage (%).

| Land-Use Change / Method | MC | LR | GAM | SA |
|---|---|---|---|---|
| Low-density residential | 15.39 | 0 | 0 | 0 |
| Medium-density residential | 40.15 | 0.29 | 6.09 | 1.10 |
| High-density residential | 0.29 | 2.78 | 4.27 | 3.17 |
| Commercial | 2.95 | 0.90 | 9.70 | 1.29 |
| Industrial | 0.01 | 0 | 0 | 0 |
| Institution | 0 | n/a | n/a | n/a |
| Transportation | 2.5 | 0.14 | 3.72 | 2.86 |
| Protected area and recreation | 17.79 | 0 | 0 | 0 |
| Agriculture | 6.65 | n/a | n/a | n/a |
| Water | 22.13 | n/a | n/a | n/a |
| Under development | 17.39 | 1.89 | 0.34 | 1.36 |
| Overall | 0.01 | 0.53 | 1.86 | 0.19 |

**Table A9.** Absolute difference between overall accuracies of final models derived from 10-fold SCV with full-balanced and reduced-balanced test datasets in percentage (%).

| Land-Use Change / Method | LR | GAM | SA |
|---|---|---|---|
| Low-density residential | 0 | 0 | 0 |
| Medium-density residential | 1.47 | 0.73 | 0.11 |
| High-density residential | 2.28 | 1.68 | 2.67 |
| Commercial | 1.06 | 8.43 | 0.95 |
| Industrial | 0 | 0 | 0 |
| Institution | n/a | n/a | n/a |
| Transportation | 2.58 | 0.65 | 1.70 |
| Protected area and recreation | 0 | 0 | 0 |
| Agriculture | n/a | n/a | n/a |
| Water | n/a | n/a | n/a |
| Under development | 0.63 | 3.08 | 0.72 |
| Overall | 0.64 | 1.40 | 0.56 |

## Appendix E

In the following context, a smoothed term refers to a variable that was fit using a smoothing function to represent the non-linear relationship between it and the response variable in GAM. [Notes: low-density residential (LDR), medium-density residential (MDR), high-density residential (HDR), commercial (COM), industrial (IND), institution (INS), transportation (TRA), protected area and recreation (REC), agriculture (AGR), water (WAT), under development (UND)]

**Table A10.** Coefficients of significant land-use change predictors in final LR derived from CCV with full-balanced test datasets.

| Predictor / Method | Final LR | | | | | | | |
|---|---|---|---|---|---|---|---|---|
| | LDR | MDR | HDR | COM | IND | TRA | REC | UND |
| lu2006_2 | 1.21 | −4.43 | | | | 2.71 | 3.37 | |
| lu2006_3 | 2.63 | −2.59 | −1.49 | | | | | |
| lu2006_4 | 1.11 | −2.93 | −0.52 | −2.47 | | 1.78 | | |
| lu2006_5 | | | −1.27 | | | | | −3.69 |
| lu2006_7 | | −1.81 | | | | −1.29 | | −3.55 |
| lu2006_8 | 1.53 | | | | | | | −2.80 |
| lu2006_9 | | | | 1.75 | | | | −2.18 |
| lu2006_11 | | −2.07 | −1.05 | | | | | −3.39 |
| lc2006_2 | 1.82 | −1.65 | | | | | | |
| lc2006_3 | | −1.50 | | | | | | |
| lc2006_4 | | −1.17 | | | | | | |
| lc2006_5 | | −1.56 | | | | | | |
| lc2006_6 | | | | −2.16 | | | | |
| lc2006_7 | | −1.92 | | | | | | |
| lc2006_8 | 2.32 | −1.52 | −1.74 | | | | | |
| lc2010_2 | | 0.78 | | | | | | 2.06 |
| lc2010_3 | | 2.39 | | −0.98 | | −1.85 | | −0.90 |
| lc2010_5 | 1.06 | 1.97 | | 0.61 | | 0.52 | 4.40 | −1.47 |
| lc2010_6 | | −1.05 | | | | | 4.45 | |
| lc2010_7 | 2.62 | 3.91 | | 0.61 | | | 6.51 | −2.86 |
| lc2010_8 | 2.89 | 1.21 | | | | 2.09 | | 1.51 |

**Table A10.** *Cont*.

| Method<br>Predictor | LDR | MDR | HDR | COM | IND | TRA | REC | UND |
|---|---|---|---|---|---|---|---|---|
| ParcelArea | | | −0.01 | −48.91 | | −40.99 | −41.70 | 5.19 |
| DA_Area | | | | −28.73 | | | | |
| MeanSlope | | | | | | | | −3.52 |
| MeanDEM | | −3.54 | 5.94 | | | −9.62 | | −9.18 |
| Wood_dist | | 0.33 | −0.70 | | | | | −0.72 |
| River_dist | | | | | | | 2.11 | |
| Water_dist | | 0.32 | | | | | | 0.52 |
| LRoad_dist | | 6.47 | | −9.23 | | 2.41 | | |
| MRoad_dist | 0.68 | | | 1.26 | | | | |
| Ramp_dist | | | | | | 0.08 | | |
| lu4_dist | −0.41 | −10.42 | −2.78 | 8.69 | | −0.86 | | −1.33 |
| lu5_dist | | | | | | | | |
| lu8_dist | 2.03 | −1.71 | 1.05 | | | −3.80 | 15.49 | |
| lu9_dist | | | −0.22 | 0.18 | | | | |
| Residential_<br>Popn_Density | −0.04 | | | −0.04 | 0.03 | | | |
| DA_Popn_<br>Density | | $-1.31 \times 10^{-4}$ | $-1.49 \times 10^{-4}$ | | | | | $2.74 \times 10^{-4}$ |
| F_lu1 | | | | | | | | 2.48 |
| F_lu2 | | | | | | | | 1.46 |
| F_lu7 | | 0.83 | | | | | | 1.25 |
| F_lu8 | | | 3.94 | | | 3.00 | | |
| F_lu9 | | | | | | | | 2.30 |
| F_lu11 | | 0.84 | | | | 0.76 | | |

**Table A11.** Coefficients of significant land-use change predictors in final LR derived from CCV with reduced-balanced test datasets.

| Method<br>Predictor | LDR | MDR | HDR | COM | IND | TRA | REC | UND |
|---|---|---|---|---|---|---|---|---|
| lu2006_2 | 1.21 | | | | | 4.04 | 3.37 | |
| lu2006_3 | 2.63 | | −1.36 | | | | | |
| lu2006_4 | 1.11 | | | −2.51 | | | | |
| lu2006_5 | | | −1.13 | | | | | −3.57 |
| lu2006_7 | | | | | | | | −2.71 |
| lu2006_8 | 1.53 | | 2.75 | | | | | −1.96 |
| lu2006_9 | | | | | | | | −2.07 |
| lu2006_11 | | | | | | | | −3.24 |
| lc2006_2 | 1.82 | −4.23 | | | | | | |
| lc2006_3 | | −2.99 | | | | | | |
| lc2006_5 | | −3.74 | | | | | | |
| lc2006_6 | | | | | | | | −5.08 |
| lc2006_7 | | −3.77 | | | | | | |
| lc2006_8 | 2.32 | −2.74 | | | | | | |
| lc2010_2 | | 1.69 | −1.42 | | | | | |
| lc2010_3 | | 3.47 | | | | −1.28 | | −1.03 |
| lc2010_5 | 1.06 | 2.33 | | | | 0.97 | 4.40 | |
| lc2010_6 | | | | | | | 4.45 | |
| lc2010_7 | 2.62 | 5.01 | | | | | 6.51 | −2.65 |
| lc2010_8 | 2.89 | | | | | 1.91 | | 2.39 |
| ParcelArea | | | −48.15 | −209.63 | | −17.55 | −41.70 | 12.70 |
| DA_Area | | | | −54.93 | | | | |
| MeanSlope | | | | | | | | −0.59 |
| Wood_dist | | | −0.69 | | | | | −0.67 |
| River_dist | | | | | | | 2.11 | |
| LRoad_dist | | 9.73 | | −6.63 | | | | |
| MRoad_dist | 0.67 | | | | | | | −1.42 |
| Ramp_dist | | 0.17 | | | | | | |
| lu4_dist | −0.41 | −6.01 | −2.72 | 19.15 | | −1.16 | | −3.04 |
| lu5_dist | | | | | | | | |
| lu8_dist | 2.03 | | | | | −3.44 | 15.49 | |
| lu9_dist | | | −0.27 | | | | | |
| Residential_<br>Popn_Density | | 0.03 | | | | | | |
| DA_Popn_<br>Density | | | $-1.43 \times 10^{-4}$ | | | | | $2.42 \times 10^{-4}$ |
| Change_<br>AveIncome | | | | −1.02 | | | | |
| F_lu1 | | | | | | | | 3.10 |
| F_lu2 | | −1.37 | | | | | | |
| F_lu7 | | −2.56 | | | | | | |
| F_lu9 | | | | −2.69 | | | | |
| F_lu11 | | | | | | | | −1.77 |

**Table A12.** Coefficients of significant land-use change predictors in final GAM derived from CCV with full-balanced test datasets.

| Method / Predictor | Final GAM | | | | | | | |
|---|---|---|---|---|---|---|---|---|
| | LDR | MDR | HDR | COM | IND | TRA | REC | UND |
| lu2006_2 | 1.22 | −3.76 | | −1.24 | | 2.08 | | |
| lu2006_3 | 2.67 | −2.66 | −1.65 | | | | | |
| lu2006_4 | 0.98 | −2.74 | −1.05 | −4.51 | | | | |
| lu2006_5 | | | −1.81 | | | | | −3.68 |
| lu2006_7 | | | | −1.46 | | −2.00 | | −3.99 |
| lu2006_8 | 1.54 | | | | | | | −3.54 |
| lu2006_9 | | | −3.44 | | | | | −3.52 |
| lu2006_11 | | | −1.32 | −2.00 | | | | −4.41 |
| lc2006_2 | 1.76 | −2.11 | | | | | | |
| lc2006_3 | | −1.94 | | | | | | |
| lc2006_4 | | −1.87 | | | | | | |
| lc2006_5 | | −1.88 | | | | | | |
| lc2006_7 | 0.66 | −1.93 | | | | | | |
| lc2006_8 | 2.61 | −1.92 | | | | | | |
| lc2010_2 | | −0.41 | −2.23 | 2.28 | | | | |
| lc2010_3 | 1.17 | | −1.09 | 1.09 | | −0.98 | | −2.53 |
| lc2010_5 | 0.96 | 1.90 | | 1.18 | | 0.82 | | −2.63 |
| lc2010_7 | 2.66 | 4.32 | | 2.65 | | 0.88 | | −4.76 |
| lc2010_8 | 2.37 | | −1.20 | 2.59 | | 1.61 | | |
| ParcelArea | | | −6.83 | S | | S | | S |
| MeanSlope | | S | | | | | | |
| MeanDEM | | | S | | | S | | S |
| Wood_dist | | S | S | | | | | S |
| River_dist | | | | | | | | S |
| Water_dist | | S | | S | | | | |
| LRoad_dist | S | S | | S | | S | | |
| MRoad_dist | S | | | S | | S | | S |
| Ramp_dist | | S | | S | | | | S |
| lu4_dist | | S | S | S | | S | | S |
| lu5_dist | | | | S | | | | |
| lu8_dist | | S | | | | S | | S |
| lu9_dist | | | S | S | | | | |
| Residential_Popn_Density | S | | S | | | | | |
| DA_Popn_Density | | S | S | | | | | |
| Change_AveIncome | | | S | S | | | | |
| Change_Popn | | S | | | | | | S |
| F_lu2 | | | | | | | | S |
| F_lu11 | | 0.69 | | | | | | |

Note: The symbol "S" in the table indicates that the predictor is considered significant as a smoothed term.

**Table A13.** Coefficients of significant land-use change predictors in final GAM derived from CCV with reduced-balanced test datasets.

| Method / Predictor | Final GAM | | | | | | | |
|---|---|---|---|---|---|---|---|---|
| | LDR | MDR | HDR | COM | IND | TRA | REC | UND |
| lu2006_2 | 1.22 | | | | | 4.56 | | 2.16 |
| lu2006_3 | 2.67 | | −1.52 | | | | | |
| lu2006_4 | 0.98 | | −0.84 | −3.79 | | | | |
| lu2006_5 | | | −1.51 | | | | | |
| lu2006_8 | 1.54 | | | | | | | |
| lu2006_9 | | | | | | | | −1.39 |
| lu2006_11 | | | | | | | | −2.07 |
| lc2006_2 | 1.76 | | | | | | | 2.48 |
| lc2006_3 | | | | | | | | 3.54 |
| lc2006_5 | | | | | | | | 2.59 |
| lc2006_7 | 0.66 | | | | | | | 2.18 |
| lc2006_8 | 2.61 | | | | | | | 2.28 |
| lc2010_2 | | | −2.18 | 4.85 | | | | |
| lc2010_3 | | 3.28 | −2.05 | 3.80 | | | | −2.41 |
| lc2010_5 | 0.96 | 2.85 | −0.82 | 3.52 | | 1.45 | | −1.25 |
| lc2010_6 | | | | | | 2.14 | | |
| lc2010_7 | 2.66 | 4.51 | | 5.52 | | 1.76 | | −3.69 |
| lc2010_8 | 2.37 | −2.34 | | 4.51 | | | | |

**Table A13.** *Cont*.

| Method | | | | | | | | |
|---|---|---|---|---|---|---|---|---|
| **Predictor** | **LDR** | **MDR** | **HDR** | **COM** | **IND** | **TRA** | **REC** | **UND** |
| ParcelArea | | | S | S | | S | | S |
| DA_Area | | | S | S | | | | |
| MeanSlope | | | | | | | | S |
| MeanDEM | | | S | | | | | S |
| Wood_dist | | | S | | | | | S |
| River_dist | | | | | | | | S |
| LRoad_dist | S | | | −9.98 | | S | | |
| MRoad_dist | S | | | | | S | | S |
| Ramp_dist | | S | | | | | | |
| lu4_dist | | S | S | S | | S | | S |
| lu5_dist | | | | S | | | | |
| lu8_dist | | | | | | S | | S |
| Residential_Popn_Density | S | | S | S | | | | S |
| DA_Popn_Density | | | | | | | | S |
| F_lu2 | | −1.81 | | −1.14 | | | | |
| F_lu11 | | | | | | | | −2.09 |

Note: The symbol "S" in the table indicates that the predictor is considered significant as a smoothed term.

**Table A14.** Coefficients of significant land-use change predictors in final SA derived from CCV with full-balanced test datasets.

| Method | | | | | | | | |
|---|---|---|---|---|---|---|---|---|
| **Predictor** | **LDR** | **MDR** | **HDR** | **COM** | **IND** | **TRA** | **REC** | **UND** |
| lu2006_2 | 0.72 | −1.91 | | | | | 1.87 | −0.85 |
| lu2006_3 | 1.38 | −0.60 | −0.81 | | | | | |
| lu2006_4 | 0.64 | −0.72 | | −1.34 | | | | −0.81 |
| lu2006_5 | | | −0.76 | | | | | −2.37 |
| lu2006_7 | | | | | 2.58 | −1.05 | 1.30 | −2.26 |
| lu2006_8 | 0.84 | | | | | | | −1.79 |
| lu2006_9 | 1.58 | | | 0.53 | | | 1.01 | −1.02 |
| lu2006_10 | | −0.52 | | | | | | |
| lu2006_11 | 0.70 | −0.35 | | | | | | −1.91 |
| lc2006_2 | 1.01 | −0.60 | | | 2.71 | | | |
| lc2006_3 | | −0.36 | | | 2.38 | | | |
| lc2006_4 | | | | | | | | −1.49 |
| lc2006_5 | | −0.49 | | | 2.26 | | −1.48 | −1.28 |
| lc2006_6 | | −0.91 | | | | | | −2.06 |
| lc2006_7 | | −0.63 | | | | | −1.16 | |
| lc2006_8 | 0.76 | −0.35 | | | | | −1.42 | −1.23 |
| lc2010_2 | | 0.75 | −1.26 | | | | | |
| lc2010_3 | | 1.63 | | −0.57 | | −1.02 | | −0.74 |
| lc2010_5 | 0.59 | 1.53 | | 0.35 | | 0.31 | 2.43 | −0.69 |
| lc2010_6 | | −0.97 | | | | | 1.88 | |
| lc2010_7 | 1.05 | 1.83 | | 0.35 | −1.42 | | 3.25 | −1.99 |
| lc2010_8 | 1.02 | 1.19 | | | | 0.59 | | |
| ParcelArea | | | −75.41 | −16.84 | | −20.50 | −19.21 | 2.05 |
| DA_Area | | | | −25.10 | | | | |
| MeanSlope | | | | | | | | −0.19 |
| MeanDEM | | −2.24 | | 2.62 | | −4.61 | | |
| Wood_dist | | 0.29 | −0.45 | | | | | |
| River_dist | | | | | | | 0.88 | |
| Water_dist | | −0.15 | | | | | | |
| LRoad_dist | | 0.73 | | −2.72 | | | | |
| MRoad_dist | | −0.28 | | | | | | −0.63 |
| Ramp_dist | 0.03 | | | | | 0.05 | | |
| lu4_dist | | −4.80 | −1.90 | 1.49 | −24.46 | | | |
| lu5_dist | | | | | | | | |
| lu8_dist | 0.53 | −0.58 | | | | −1.85 | 3.07 | |
| lu9_dist | | | | | | | | |
| Residential_Popn_Density | | −0.02 | −0.02 | 0.02 | | | | |
| DA_Popn_Density | | $-0.60 \times 10^{-4}$ | $-1.16 \times 10^{-4}$ | | | | | $1.07 \times 10^{-4}$ |
| Change_AveIncome | | −0.20 | | | | | | |
| Change_Popn | | −0.03 | | | −0.61 | | | |
| F_lu1 | | | | | | | | 0.79 |
| F_lu2 | | | | | | | | 0.42 |
| F_lu3 | | | | | | | | |
| F_lu7 | | | 0.54 | 0.52 | 1.40 | | | |
| F_lu9 | | | | | 0.97 | | | |
| F_lu11 | | | | | | | | |

**Table A15.** Coefficients of significant land-use change predictors in final SA derived from CCV with reduced-balanced test datasets.

| Method / Predictor | Final SA | | | | | | | |
|---|---|---|---|---|---|---|---|---|
| | LDR | MDR | HDR | COM | IND | TRA | REC | UND |
| lu2006_2 | 0.72 | −2.09 | | | | 0.91 | 1.87 | −0.67 |
| lu2006_3 | 1.38 | | −0.85 | | | | | |
| lu2006_4 | 0.64 | −1.04 | | −1.16 | | | | |
| lu2006_5 | | | −1.08 | | | | | −2.58 |
| lu2006_7 | | | 0.88 | | 2.58 | −0.99 | 1.30 | |
| lu2006_8 | 0.84 | | 0.92 | | | | | −0.66 |
| lu2006_9 | 1.59 | | | | | | 1.01 | −2.11 |
| lu2006_11 | 0.70 | | | | | | | −1.96 |
| lc2006_2 | 1.01 | | | | 2.71 | | | |
| lc2006_3 | | | | | 2.38 | | | |
| lc2006_5 | | | | | 2.26 | | −1.48 | |
| lc2006_6 | | | | | | | | −1.63 |
| lc2006_7 | | | | | | | −1.16 | |
| lc2006_8 | 0.76 | | | | | | −1.42 | |
| lc2010_2 | | 1.42 | −1.19 | | | | | |
| lc2010_3 | | 2.42 | | −0.71 | | −0.76 | | −0.62 |
| lc2010_4 | | | | | | | | |
| lc2010_5 | 0.59 | 1.86 | | | | 0.62 | 2.43 | |
| lc2010_6 | | | | | | | 1.88 | |
| lc2010_7 | 1.06 | 2.12 | | | −1.42 | | 3.25 | −0.91 |
| lc2010_8 | 1.02 | 0.96 | −1.00 | | | 0.57 | | |
| ParcelArea | | | −49.28 | −105.50 | | | −19.21 | |
| MeanSlope | | | −0.13 | | | | | |
| MeanDEM | | | | | | −3.85 | | |
| Wood_dist | | | −0.35 | | | | | |
| River_dist | | −0.42 | | | | | 0.88 | |
| MRoad_dist | | | | | | | | −1.16 |
| Ramp_dist | 0.03 | | | | | | | |
| lu4_dist | | −3.13 | −1.22 | 2.00 | −24.46 | −1.10 | | 0.47 |
| lu5_dist | | | | 0.81 | | | | |
| lu8_dist | 0.53 | | | | | −1.36 | 3.07 | |
| lu9_dist | | | −0.27 | | | | | |
| Residential_Popn _Density | | | −0.02 | | | | | |
| DA_Popn_Density | | | | | | | | $1.21 \times 10^{-4}$ |
| Change_Popn | | | | | −0.61 | | | |
| Change_AveIncome | | | | −0.46 | | | | |
| F_lu7 | | | | | 1.40 | | | 0.81 |
| F_lu9 | | | | | 0.97 | | | |

**Table A16.** Coefficients of significant land-use change predictors in final LR derived from SCV with full-balanced test datasets.

| Method / Predictor | Final LR | | | | | | | |
|---|---|---|---|---|---|---|---|---|
| | LDR | MDR | HDR | COM | IND | TRA | REC | UND |
| lu2006_2 | 1.49 | −4.36 | −1.54 | | | 2.89 | 3.51 | |
| lu2006_3 | 3.87 | −2.43 | −0.62 | | | 1.90 | | |
| lu2006_4 | 1.74 | −2.80 | −1.14 | −2.43 | | | | |
| lu2006_5 | | | | | | | | −4.07 |
| lu2006_7 | | −1.55 | | | | −1.27 | | −3.75 |
| lu2006_8 | 1.57 | | | | | | | −3.08 |
| lu2006_9 | | | | | 2.23 | | | −1.81 |
| lu2006_11 | 1.49 | −2.00 | −1.17 | | | | 1.66 | −3.68 |
| lc2006_2 | | −1.65 | 1.41 | | | | | |
| lc2006_3 | | −1.42 | | | | | | |
| lc2006_4 | | −1.11 | | | | | | |
| lc2006_5 | | −1.50 | | | | | | |
| lc2006_6 | | | | | | | 4.35 | |
| lc2006_7 | | −1.82 | | | | | 4.37 | |
| lc2006_8 | 1.06 | −1.38 | | | | | 6.39 | |
| lc2010_2 | | 0.81 | | | | | | 2.00 |
| lc2010_3 | | 2.48 | | −0.98 | | −1.84 | −3.88 | −0.90 |
| lc2010_5 | 0.77 | 2.04 | | 0.54 | | 0.53 | | −1.47 |
| lc2010_6 | 0.59 | −1.00 | | 1.90 | | | | |
| lc2010_7 | 1.08 | 3.88 | | 0.61 | | | | −2.69 |
| lc2010_8 | 1.67 | 1.33 | | | | 2.16 | 2.30 | 1.12 |

**Table A16.** *Cont*.

| Method / Predictor | LDR | MDR | HDR | COM | IND | TRA | REC | UND |
|---|---|---|---|---|---|---|---|---|
| ParcelArea | | | −110.10 | −60.91 | | −43.29 | | 5.03 |
| DA_Area | 24.30 | | | −30.89 | | 30.50 | | |
| MeanSlope | −0.21 | | | | | | | −0.34 |
| MeanDEM | | | | | | −9.38 | | −10.43 |
| Wood_dist | | 0.37 | −0.86 | | | | 15.36 | −1.12 |
| Water_dist | | −0.42 | | | | | | 0.72 |
| LRoad_dist | | 6.21 | | −1.00 | | 2.67 | | |
| MRoad_dist | 0.76 | | | 1.38 | | | | |
| Ramp_dist | | | | | | 0.07 | | |
| lu4_dist | −0.42 | −10.28 | −3.07 | 10.34 | −38.12 | −1.14 | | |
| lu5_dist | | 0.45 | | | 24.05 | | | |
| lu8_dist | | −1.79 | 1.46 | | | −3.90 | | |
| lu9_dist | | | | 0.26 | | | | 0.54 |
| Residential_Popn_Density | | −0.03 | −0.03 | 0.03 | | | | |
| DA_Popn_Density | | $-1.32 \times 10^{-4}$ | $-1.62 \times 10^{-4}$ | | | | | $2.40 \times 10^{-4}$ |
| Change_Popn | | −0.18 | | | | | | |
| F_lu1 | | −0.77 | | | | | | 1.77 |
| F_lu2 | | | | | | | | 1.24 |
| F_lu7 | | 0.78 | | | | | | |
| F_lu8 | | | −4.02 | | | 2.79 | | |
| F_lu11 | | | | | | 0.94 | | |

**Table A17.** Coefficients of significant land-use change predictors in final LR derived from SCV with reduced-balanced test datasets.

| Method / Predictor | LDR | MDR | HDR | COM | IND | TRA | REC | UND |
|---|---|---|---|---|---|---|---|---|
| lu2006_2 | 1.49 | −3.45 | | | | | 3.51 | |
| lu2006_3 | 3.87 | | −1.30 | | | | | |
| lu2006_4 | 1.74 | | | −2.73 | | | | |
| lu2006_5 | | | −1.11 | | | | | −3.21 |
| lu2006_7 | | | | | | | | −2.50 |
| lu2006_8 | 1.57 | | | | | | | −1.89 |
| lu2006_9 | | | | | | | | −2.00 |
| lu2006_11 | 1.49 | | | | | | 1.66 | −2.89 |
| lc2006_6 | | | | | | | | −4.75 |
| lc2006_8 | 1.06 | | | | | | | |
| lc2010_2 | | 1.34 | −1.49 | 1.10 | | | | 1.39 |
| lc2010_3 | | 3.49 | | | | | | −1.16 |
| lc2010_5 | 0.77 | 1.89 | | | | | 4.35 | |
| lc2010_6 | 0.59 | | | | | | 4.37 | |
| lc2010_7 | 1.08 | 3.56 | | 0.93 | | | 6.39 | −2.72 |
| lc2010_8 | 1.67 | | | 1.39 | | | | 2.41 |
| ParcelArea | 24.30 | | −46.73 | −45.64 | | | −3.88 | 12.38 |
| MeanSlope | −0.21 | | | 0.41 | | | | −0.63 |
| MeanDEM | | | | 9.70 | | | | |
| Wood_dist | | | −0.92 | | | | | |
| River_dist | | | | | | | 2.30 | |
| LRoad_dist | | | | −7.23 | | | | |
| MRoad_dist | 0.76 | | | | | | | |
| lu4_dist | −0.42 | −4.87 | −3.02 | 13.83 | −38.12 | −2.05 | | −3.27 |
| lu5_dist | | | | | 24.05 | | | |
| lu8_dist | | | | | | −3.06 | 15.36 | |
| lu9_dist | | | | 0.33 | | | | |
| Residential_Popn_Density | | | | | | −0.04 | | |
| DA_Popn_Density | | | $-1.64 \times 10^{-4}$ | | | | | $2.44 \times 10^{-4}$ |
| Change_AveIncome | | | | −1.19 | | | | |
| F_lu1 | | | | | | | | 3.91 |
| F_lu2 | −1.52 | | | | | | | |
| F_lu8 | | | | −3.91 | | | | |
| F_lu11 | | | | | | | | −1.85 |

**Table A18.** Coefficients of significant land-use change predictors in final GAM derived from SCV with full-balanced test datasets.

| Method / Predictor | Final GAM | | | | | | | |
|---|---|---|---|---|---|---|---|---|
| | LDR | MDR | HDR | COM | IND | TRA | REC | UND |
| lu2006_2 | | −3.96 | −1.87 | −1.54 | | | | |
| lu2006_3 | 2.90 | −3.13 | −1.19 | | | | | |
| lu2006_4 | 1.00 | −2.91 | −1.98 | −4.79 | | | | |
| lu2006_5 | | | | | | | | −2.87 |
| lu2006_7 | | | | | | −2.77 | | −3.32 |
| lu2006_8 | 1.74 | | | | | | | −2.28 |
| lu2006_9 | | | −3.52 | | | | | −2.42 |
| lu2006_11 | 1.41 | | −2.06 | −2.32 | | | | −3.03 |
| lc2006_2 | 1.50 | −1.86 | | | | | | |
| lc2006_3 | 1.46 | −1.58 | | | | | | |
| lc2006_4 | 0.91 | −1.55 | | | | | | |
| lc2006_5 | | −1.64 | | | | | | |
| lc2006_7 | | −1.70 | | | | | | |
| lc2006_8 | 1.28 | −1.60 | | | | | | |
| lc2010_2 | | −0.69 | | 2.43 | | | | |
| lc2010_3 | | 0.90 | | 1.11 | | −1.02 | | −2.31 |
| lc2010_5 | 0.92 | 1.80 | | 1.17 | | 0.78 | | −2.38 |
| lc2010_7 | 0.58 | 4.11 | | 2.59 | | 0.78 | | −3.98 |
| lc2010_8 | 1.11 | −0.63 | −1.22 | 2.75 | | 1.58 | | |
| ParcelArea | | | −73.74 | S | | S | | |
| DA_Area | 35.23 | S | | | | | | |
| MeanSlope | | S | | | | | | |
| MeanDEM | | | | | | S | | |
| Wood_dist | | S | S | S | | | | S |
| Water_dist | | S | | | | | | |
| LRoad_dist | S | S | | S | | S | | S |
| MRoad_dist | S | | | S | | S | | S |
| Ramp_dist | | | | S | | | | |
| lu4_dist | −0.39 | S | S | S | | S | | S |
| lu5_dist | | | | S | | | | |
| lu8_dist | | S | | | | S | | S |
| lu9_dist | | | | S | | | | |
| Residential_Popn_Density | | S | S | | | | | |
| DA_Popn_Density | | | S | S | | | | |
| Change_AveIncome | | | S | S | | | | |
| Change_Popn | | S | | S | | | | S |
| F_lu2 | | | | | | | | S |

Note: The symbol "S" in the table indicates that the predictor is considered significant as a smoothed term.

**Table A19.** Coefficients of significant land-use change predictors in final GAM derived from SCV with reduced-balanced test datasets.

| Method / Predictor | Final GAM | | | | | | | |
|---|---|---|---|---|---|---|---|---|
| | LDR | MDR | HDR | COM | IND | TRA | REC | UND |
| lu2006_2 | | −4.56 | −1.45 | | | 5.00 | | |
| lu2006_3 | 2.90 | | −0.89 | | | | | |
| lu2006_4 | 1.00 | −4.81 | −1.51 | −3.95 | | | | |
| lu2006_5 | | | | | | | | −3.36 |
| lu2006_7 | | | | | | | | −2.44 |
| lu2006_8 | 1.74 | | | -2.50 | | | | |
| lu2006_9 | | | | | | | | −2.55 |
| lu2006_11 | 1.41 | | | | | | | −3.38 |
| lc2006_2 | 1.50 | | | | | | | |
| lc2006_3 | 1.46 | | | | | | | |
| lc2006_4 | | | | | | | | |
| lc2006_5 | 0.91 | | | | | | | |
| lc2006_6 | | | | | | | | −5.49 |
| lc2006_7 | | | | | | | | |
| lc2006_8 | 1.28 | | | | | | | |
| lc2010_2 | | | −2.06 | 3.08 | | | | |
| lc2010_3 | | 2.68 | −1.98 | 1.46 | | | | |
| lc2010_5 | 0.92 | 2.18 | −0.68 | 1.63 | | | | −1.83 |
| lc2010_6 | 0.58 | | | | | | | −1.08 |
| lc2010_7 | 1.11 | 4.27 | | 2.66 | | 1.96 | | −3.49 |
| lc2010_8 | | | | 3.12 | | 2.89 | | 2.18 |
| ParcelArea | | | S | | | S | | S |
| DA_Area | 35.23 | | S | | | | | |

**Table A19.** *Cont*.

| Method / Predictor | | | | Final GAM | | | | |
|---|---|---|---|---|---|---|---|---|
| **Predictor** | **LDR** | **MDR** | **HDR** | **COM** | **IND** | **TRA** | **REC** | **UND** |
| MeanSlope | | | | | | | | −0.64 |
| MeanDEM | | | S | | | | | |
| Wood_dist | | | −2.01 | | | | | |
| River_dist | | | | | | | | S |
| Water_dist | | S | | | | S | | |
| LRoad_dist | S | | | | | S | | |
| MRoad_dist | S | | | | | S | | |
| Ramp_dist | | | | | | | | |
| lu4_dist | −0.39 | S | S | S | | | | S |
| lu5_dist | | | | | | | | |
| lu8_dist | | | | S | | S | | S |
| lu9_dist | | | | | | | | |
| Residential_Popn_Density | | | S | | | S | | S |
| DA_Popn_Density | | | | | | | | S |
| Change_AveIncome | | | | S | | | | |
| Change_Popn | | | | | | | | |
| F_lu1 | | S | | | | | | |
| F_lu2 | | | | | | | | |
| F_lu11 | | | | | | | | |

Note: The symbol "S" in the table indicates that the predictor is considered significant as a smoothed term.

**Table A20.** Coefficients of significant land-use change predictors in final SA derived from SCV with full-balanced test datasets.

| Method / Predictor | | | | Final SA | | | | |
|---|---|---|---|---|---|---|---|---|
| **Predictor** | **LDR** | **MDR** | **HDR** | **COM** | **IND** | **TRA** | **REC** | **UND** |
| lu2006_2 | 0.90 | −1.92 | −0.84 | | 2.36 | | 1.91 | −1.16 |
| lu2006_3 | 1.33 | −0.60 | −0.25 | | | | | −1.15 |
| lu2006_4 | 0.55 | −0.73 | −0.76 | −1.37 | | | | −2.56 |
| lu2006_5 | 1.30 | | | | | | | −2.62 |
| lu2006_6 | | | | | 7.33 | | | −1.96 |
| lu2006_7 | | | | | 2.99 | −1.17 | 1.36 | −0.99 |
| lu2006_8 | 0.63 | | | | | | | −2.08 |
| lu2006_9 | 1.56 | | | 0.51 | 1.66 | | 1.07 | |
| lu2006_10 | | −0.46 | | | | | | |
| lu2006_11 | 0.48 | −0.34 | | | | | 0.80 | |
| lc2006_2 | 1.28 | −0.64 | | | 2.83 | | −1.72 | |
| lc2006_3 | 0.96 | −0.40 | | | 2.44 | | −1.35 | |
| lc2006_5 | | −0.52 | | | 2.23 | | −1.58 | |
| lc2006_6 | | −0.92 | | | | | | −1.90 |
| lc2006_7 | 0.45 | −0.68 | | | | | −1.25 | |
| lc2006_8 | 0.87 | −0.38 | | | 2.03 | | −1.57 | |
| lc2010_2 | | 0.80 | −1.42 | | | | | |
| lc2010_3 | | 1.66 | | −0.68 | | −1.01 | 3.75 | −0.71 |
| lc2010_5 | 0.67 | 1.56 | | | | 0.34 | 2.37 | −0.66 |
| lc2010_6 | −0.50 | −1.07 | | | | | 1.83 | |
| lc2010_7 | 1.24 | 1.85 | −0.31 | 0.24 | −1.10 | | 3.22 | −1.88 |
| lc2010_8 | 1.18 | 1.22 | −0.75 | | | 0.63 | | |
| ParcelArea | 8.13 | | −6.77 | −20.99 | 6.73 | −24.66 | −20.11 | |
| DA_Area | 1.56 | | | −21.53 | | | | |
| MeanSlope | | | | | | | | −0.18 |
| MeanDEM | | −2.13 | | | | | | |
| Wood_dist | | 0.28 | −0.51 | | 0.92 | | | |
| River_dist | | | | | | | 0.91 | |
| Water_dist | | −0.15 | | | | | | |
| LRoad_dist | | 0.58 | | −3.73 | | 1.45 | | |
| MRoad_dist | 0.23 | −0.26 | | 0.43 | | | | −0.62 |
| Ramp_dist | | | | | | 0.04 | | |
| lu4_dist | −0.35 | −4.65 | −2.35 | 2.53 | −25.33 | −0.52 | | |
| lu5_dist | | | | | 0.48 | | | |
| lu8_dist | 0.77 | −0.59 | | −0.85 | −1.69 | −1.96 | 2.68 | 0.81 |
| lu9_dist | | | | | 0.14 | | | |
| Residential_Popn_Density | | −0.02 | −0.02 | 0.02 | | | | |
| DA_Popn_Density | | $-0.65 \times 10^{-4}$ | $-1.05 \times 10^{-4}$ | | | | | $9.30 \times 10^{-5}$ |
| Change_AveIncome | | 0.20 | | | | | | |
| Change_Popn | | −0.03 | | | | | | |
| F_lu2 | | | | | −0.39 | | | 0.31 |
| F_lu3 | | | | | | 1.07 | | |
| F_lu7 | 1.00 | | | | | | | |
| F_lu11 | | | −0.82 | | | 0.37 | | |

**Table A21.** Coefficients of significant land-use change predictors in final SA derived from SCV with reduced-balanced test datasets.

| Predictor | Final SA | | | | | | | |
|---|---|---|---|---|---|---|---|---|
| **Method** | **LDR** | **MDR** | **HDR** | **COM** | **IND** | **TRA** | **REC** | **UND** |
| lu2006_2 | 0.90 | −2.09 | | | 2.36 | 1.05 | 1.91 | |
| lu2006_3 | 1.33 | | −0.75 | | | | | |
| lu2006_4 | 0.55 | −1.04 | | −1.29 | | | | |
| lu2006_5 | 1.30 | | | | | | | −1.88 |
| lu2006_6 | | | | | 7.33 | | | |
| lu2006_7 | | | 1.10 | | 2.99 | −0.88 | 1.36 | −1.47 |
| lu2006_8 | 0.63 | | 1.78 | | | | | −1.00 |
| lu2006_9 | 1.56 | | 3.04 | | 1.66 | | 1.07 | −1.13 |
| lu2006_11 | 0.48 | | | | | | 0.80 | −1.47 |
| lc2006_2 | 1.28 | | | | 2.83 | | −1.72 | |
| lc2006_3 | 0.96 | | | | 2.44 | | −1.35 | |
| lc2006_5 | | | | | 2.23 | | −1.58 | |
| lc2006_7 | 0.45 | | | | | | −1.25 | |
| lc2006_8 | 0.87 | | | | 2.03 | | −1.57 | |
| lc2010_2 | | 1.42 | −1.04 | | | | | |
| lc2010_3 | | 2.42 | | −0.77 | | −0.73 | 3.75 | −0.49 |
| lc2010_5 | 0.67 | 1.86 | | | | 0.60 | 2.37 | −0.39 |
| lc2010_6 | −0.50 | | | | | | 1.83 | |
| lc2010_7 | 1.24 | 2.12 | | | −1.10 | | 3.22 | −1.82 |
| lc2010_8 | 1.18 | 0.96 | | | | | | |
| ParcelArea | 8.13 | | −39.48 | −101.00 | 6.73 | | −20.11 | 6.14 |
| DA_Area | 1.56 | | | | | | | |
| MeanSlope | | | | | | | | −0.20 |
| Wood_dist | | | −0.60 | | 0.92 | | | |
| Water_dist | | | 0.26 | | | | | |
| River_dist | | −0.42 | | | | | 0.91 | |
| LRoad_dist | | | | | | | | 0.84 |
| MRoad_dist | 0.23 | | | | | | | −0.79 |
| lu4_dist | −0.35 | −3.13 | −2.20 | 2.06 | −25.33 | −1.46 | | −1.29 |
| lu5_dist | | | | 0.53 | 0.48 | | | −1.18 |
| lu8_dist | 0.77 | | | | −1.69 | −1.14 | 2.68 | |
| DA_Popn_Density | | | $-1.21 \times 10^{-4}$ | | | | | |
| F_lu1 | | | | −1.70 | | | | |
| F_lu2 | | | | | −0.39 | | | |
| F_lu7 | 1.00 | | | | | | | |
| F_lu11 | | | | | | | | −2.10 |

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
