# Peer review of "Comparison of Statistical Approaches for Modelling Land-Use Change"

_land, doi:10.3390/land7040144_

Round 1

Reviewer 1 Report

The authors used three different modeling approaches to test their ability to model land use change. I found the paper very interesting and well written. I do, however, have some concerns that I'd like to see addressed before recommending this paper for publication. In detail, these are:

- In the Introduction, the authors mention this is the first paper to do such analysis, but I disagree with the statement, at least if you consider other types of land cover change, such as tropical deforestation.

- Given the rise of machine learning approaches (e.g., random forests) in the literature, I was wondering why the authors did not test one of these methods. These have the potential to be much better at predicting changes. 

- Line 73: km2 should be a superscript

- Line 81: there's an empty space missing

- The authors did not introduce or discuss the selection of the predictors. Given that all these models rely heavily on the input data (garbage in, garbage out rule), it is important to explain to the reader why these predictors were selected, where the data comes from, and what are the potential biases that might arise from using these. 

- the authors also do not acknowledge that there are mixed approaches that use both MCMC and logistic regression combined. See for example,

- Line 153: what is a roulette-wheel approach?

- Line 168: In my opinion, one of the problems with the GAMs is that they are also much less interpretable than other modeling approaches. Perhaps something that could be mentioned.

- Line 205: how was the balanced dataset obtained? It was not clear to me.

- Tables: I suggest using the full name of the land use classes, rather than the abbreviations

- In the Methods section, there are way too many acronyms making the text very hard to read. I'd suggest getting rid of some of the acronyms to improve legibility. 

- Lines 303-304: should be part of the Discussion, not the Results

- Lines 321-324: should be part of the Discussion, not the Results

- Throughout the Results and Discussion, it is not clear if there was any variable selection procedure for each of the methods, and if so, which variables were selected to be in the final model. Do these differ depending on the method employed? What are the similarities among methods regarding the most important variables?

- my main concern is that the authors never differentiate the ability of the model(s) to predict the rate and/or location of change. Were these assessed differently? And if so, how and what are the main findings?

- another important concern is the issue of spatial and temporal scale. Were the models tested at different scales? I'm wondering if there are some methodologies better able to represent regional dynamics versus others that are better able to represent local dynamics. In addition, how about the abilit over time?

Author Response

Thank for providing valuable comments. Please see the attached word document for point-by-point response to these comments.

Reviewer 2 Report

Overall, the study is well conceived and well written. I have some minor suggestions:

If the paper is about land use change, there is no point of using the term land-cover or LUCC.

Introduction:

The introduction is written well. However, I would suggest the authors to include some examples. For instance, they described the benefits and trade-offs of statistical approach over ABM, they can include an example here. They also said that there is a lack of review on the comparison of multiple statistical approaches. They should include some examples here as well. They should do some literature review and tell us where is the research gap by discussing the existing literature briefly.  

Line 53-54: Unclear. Rephrase or erase, please.

Line 68: The first research question is confusing. Do you want to know the overall accuracy of the four statistical approaches you are testing? Or, the accuracy of each approach? If it's overall CCV approach, be clear about it.

Methodology:

Well written, looks fine to me.

Results:

Okay

Discussion:

Line 392-401: This paragraph should be the first paragraph in the 'Conclusion' section.

Section 4.3.: This section comes from nowhere. The authors can rename this part as 'Future direction' and instead of just focusing on ABM, they can broadly discuss to which direction the LUChange modeling going to. 

Conclusion:

Okay

Author Response

(The authors gave the same response as above.)
